# Diffusion Beats Autoregressive in Data-Constrained Settings

**Mihir Prabhudesai**[*]
Carnegie Mellon University

**Mengning Wu**[*]
Carnegie Mellon University

**Amir Zadeh**
Lambda

**Katerina Fragkiadaki**
Carnegie Mellon University

**Deepak Pathak**
Carnegie Mellon University

## Abstract

Autoregressive (AR) models have long dominated the landscape of large language models, driving progress across a wide range of tasks. Recently, diffusion-based language models have emerged as a promising alternative, though their advantages over AR models remain underexplored. In this paper, we systematically study masked diffusion models in data-constrained settings—where training involves repeated passes over limited data—and find that they significantly outperform AR models when compute is abundant but data is scarce. Diffusion models make better use of repeated data, achieving lower validation loss and superior downstream performance. We find new scaling laws for diffusion models and derive a closed-form expression for the critical compute threshold at which diffusion begins to outperform AR. Finally, we explain why diffusion models excel in this regime: their randomized masking objective implicitly trains over a rich distribution of token orderings, acting as an implicit data augmentation that AR's fixed left-to-right factorization lacks. Our results suggest that when data, not compute, is the bottleneck, diffusion models offer a compelling alternative to the standard AR paradigm. Our code is available at: `https://diffusion-scaling.github.io`.

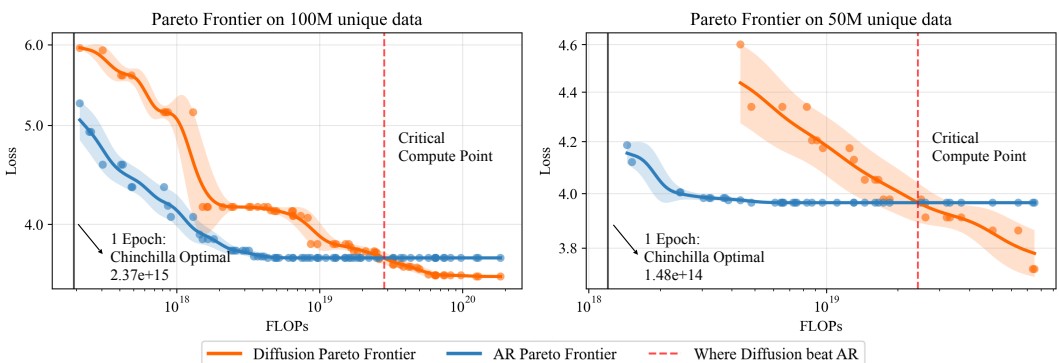

Figure 1: Pareto frontier of validation loss versus training FLOPs for autoregressive (AR) and masked diffusion models under data-constrained settings. Each point represents a model trained until convergence; we report the best validation loss achieved among all models using less than or equal to the FLOPs shown on the x-axis. AR models initially outperform diffusion models, particularly near the Chinchilla-optimal compute point [12] (indicated on the plot). However, as training continues beyond this regime with repeated data, AR models quickly saturate and begin to overfit. In contrast, diffusion models continue to improve with more compute and exhibit no signs of overfitting.

---

[*]Project co-leads & Equal contribution. Correspondence to `{mprabhud,mengninw}@andrew.cmu.edu`.

39th Conference on Neural Information Processing Systems (NeurIPS 2025).

# 1 Introduction

Training large language models (LLMs) on massive corpora of internet text has become the driver of recent AI breakthroughs [4, 28, 40]. This progress has been fueled by scaling two core resources: compute and data [15, 11]. While compute availability is steadily growing—enabled by advances in hardware and the construction of larger data centers—the growth in high-quality data is stagnating. Recent projections, such as those by Villalobos et.al. [42], estimate that the global supply of publicly available, human-generated data may be exhausted in the coming years, posing a serious bottleneck to further scaling. This looming constraint makes it increasingly important to develop modeling strategies that are more sample-efficient. Furthermore, there are several domains, such as robotics and healthcare, where the data, not compute, is a scarce resource even to begin with.

LLM development has so far been dominated by autoregressive (AR) models, which factorize the joint distribution of text in a fixed left-to-right order. While this modeling approach has delivered state-of-the-art performance across a range of benchmarks, it remains unclear whether it is the optimal strategy going forward. Recently, diffusion-based models—specifically masked diffusion models [2, 34, 18, 39, 1]—have emerged as an alternative strategy, framing text generation as an iterative masked denoising process rather than next-token prediction. At each step, the model predicts a randomly masked subset of tokens conditioned on the remaining ones, implicitly averaging over many conditional prediction orders instead of committing to one. Although these models have demonstrated similar scaling behavior to AR models [22, 39], their practical benefits have, so far, been modest—largely due to their high training compute requirements.

This high compute demand has become the central obstacle to wider adoption of diffusion-based language models. As noted by Nie *et al.* [22] and Swerdlow *et al.* [39], masked diffusion models require up to 16× more compute than AR models to match validation NLL—a clear disadvantage for most applications.

But a critical nuance is often overlooked: these comparisons are based entirely on single-epoch training, where each token is seen only once. This conflates compute efficiency with sample efficiency, making it unclear whether diffusion models truly need 16× more compute—or simply 16× more data.

To resolve this ambiguity, we systematically study masked diffusion models in data-constrained settings, where repeated training on limited data is the norm rather than the exception. We find that under such regimes, diffusion models substantially outperform autoregressive models across a variety of data scales and compute budgets. We train hundreds of models spanning multiple orders of magnitude in model size, data quantity, and number of training epochs to fit scaling laws for diffusion models in the data-constrained setting. We summarize some of our key findings below.

1. **Diffusion models surpass autoregressive models given sufficient compute.** Across a wide range of unique token budgets, we observe a consistent trend: autoregressive models initially outperform diffusion models at low compute, but quickly saturate. Beyond a critical compute threshold, diffusion models continue improving and ultimately achieve better performance (Section 3.1)

2. **Diffusion models benefit far more from repeated data.** Prior work [21] showed that repeating the dataset up to 4 epochs is nearly as effective as using fresh data for autoregressive models. In contrast, we find that diffusion models can be trained on repeated data for up to **100 epochs**, while having repeated data almost as effective as fresh data (Section 3.2).

3. **Diffusion models have a much higher effective epoch count.** Muennighoff *et al.* [21] fit scaling laws for AR models in data-constrainted settings and define $R_D^*$ as a learned constant that characterizes the number of epochs after which training more epochs results in significantly diminished returns. For autoregressive models, they estimate $R_D^* \approx 15$ . In contrast, we find $R_D^* \approx 500$ for diffusion models, suggesting they can benefit from repeated data over far more epochs without major degradation (Section 3.2).

4. **Critical compute point follows a power law with dataset size.** We find that the amount of compute required for diffusion models to outperform autoregressive models—the critical compute point—scales as a power law with the number of unique tokens. This yields a closed-form expression that predicts when diffusion becomes the favorable modeling choice for any given dataset size (Section 3.3).

5. **Diffusion models yield better downstream performance.** We find the above benefits extend beyond validation loss: the best diffusion model trained in data-constrained settings consistently outperform the best autoregressive model on a range of downstream language tasks (Section 3.4).

6. **Exposure to different token orderings helps explain diffusion's sample efficiency.** By adding explicit data augmentations to AR training, we find that diffusion models' advantage arises from their exposure to a diverse set of token orderings. Essentially, the randomized masking in diffusion's objective serves as implicit data augmentation, allowing it to generalize beyond the fixed left-to-right factorization of AR models. (Section 3.5)

Through detailed scaling law analysis and downstream task evaluations, we demonstrate that diffusion models make significantly better use of repeated data, achieving lower validation loss and better generalization to downstream tasks. These results suggest that diffusion models may offer a compelling and underappreciated advantage in scenarios where data—not compute—is the primary bottleneck.

## 2 Method

Our objective is to determine whether masked diffusion language models are more effective than standard autoregressive models in data-constrained settings. For studying this, we keep the core architecture and data pipeline fixed across both families.

### 2.1 Preliminaries:

**Autoregressive models.** In Autoregressive LLMs [40, 28, 4] each token is predicted based on a growing prefix of prior tokens, defining a left-to-right factorization of the sequence probability:

$$p_{\text{AR}}(x_1, \ldots, x_L) = \prod_{j=1}^{L} p(x_j \mid x_{<j}).$$

This structure is implemented using a *causal attention mask*, which prevents each token from attending to future positions. The model is trained via next-token prediction over clean, uncorrupted sequences.

**Diffusion models.** Masked diffusion language models [2, 34, 22, 39] treat generation as iterative denoising. For each training sequence $x = (x_1, \ldots, x_L)$ diffusion models

1. Corrupt the sequence by sampling a masking ratio $r \sim \mathcal{U}(0, 1)$ and independently replacing each token with a special [MASK] symbol with probability $r$. This yields a corrupted sequence $\tilde{x}$ and a mask set

$$\mathcal{M} = \{\, i \in [1, L] : \tilde{x}_i = [\text{MASK}] \,\}.$$

2. Denoise by predicting the original tokens at the masked positions with full (bidirectional) attention over $\tilde{x}$:

$$p_{\text{Diffusion}}(x \mid \tilde{x}) = \prod_{i \in \mathcal{M}} p_\theta(x_i \mid \tilde{x}).$$

Because the mask pattern is resampled for every example, the model is implicitly trained on a vast collection of token–ordering tasks. The absence of a causal mask allows each prediction to attend to *both* past and future unmasked tokens.

### 2.2 Modeling Details for AR and Masked Diffusion

Both model families share the same Transformer backbone (GPT-2 style with rotary positional embeddings, RoPE [38]).

Given a clean input sequence $x = (x_1, \ldots, x_L) \in \mathcal{V}^L$, both models minimize a token-level cross-entropy loss, yet they differ in the conditioning context:

**Autoregressive (AR) objective.** AR models predict each token conditioned on its prefix using a causal attention mask:

$$\mathcal{L}_{\text{AR}} = -\sum_{j=1}^{L} \log p_\theta\big(x_j \mid x_{<j}\big).$$

**Masked Diffusion objective.** For masked diffusion we first sample a masking ratio $r \sim \mathcal{U}(0,1)$ and construct a corrupted sequence $\tilde{x}$ by independently replacing each token with [MASK] with probability $r$. Let $\mathcal{M} = \{\, i : \tilde{x}_i = [\text{MASK}] \,\}$ be the set of masked positions. The loss is then

$$\mathcal{L}_{\text{Diffusion}} = -\mathbb{E}_{r, \tilde{x} \sim q_r} \frac{1}{r} \sum_{i \in \mathcal{M}} \log p_\theta(x_i \mid \tilde{x}),$$

which can be interpreted as an evidence lower bound (ELBO) on the data log-likelihood, and thus the loss provides an upper bound on the true negative log-likelihood.

Beyond the attention mechanism and input corruption, *all* other variables are held constant. We follow the hyperparameter configuration proposed by Muennighoff *et al.* [21] for all training runs. In particular, we use a dynamic learning rate schedule that adapts to the number of training epochs. For more detailed information on both model families please refer to related work Section in Appendix 6

## 2.3 Scaling Framework in Data-Constrained Settings

Classical scaling laws, such as those proposed by [15, 12], model validation loss as a function of total parameters ($N$) and training tokens ($D$), assuming all data is unique that is single epoch regime.

Muennighoff *et al.*[21] extend the Chinchilla framework to explicitly account for repeated data — a common necessity in data-constrained regimes. They show that repeating training data beyond a few epochs yields diminishing returns and propose a new scaling law that incorporates the decaying utility of repeated tokens.

We briefly outline their formulation below. Let $U$ denote the number of unique tokens, $E$ the number of epochs (how many times each unique token is reused), and $D = U \cdot E$ the total number of tokens seen during training.

To model diminishing returns from repeated data, Muennighoff *et al.* [21] introduce an *effective unique data size* $D'$, motivated by the idea that each additional epoch contributes less useful signal than the previous. Specifically, they assume the value extracted from the $k^{\text{th}}$ exposure to the same data follows a geometric progression, where the utility of a token on its $k$-th repetition is $(1-\delta)^{k-1}$. Summing over all epochs the total effective data becomes: $D' = U \cdot \sum_{k=1}^{E}(1-\delta)^{k-1} = U \cdot \frac{1-(1-\delta)^E}{\delta}$ where $\delta$ is the decay factor. Defining $R_D^\star = \frac{1-\delta}{\delta}$, the expression simplifies to the exponential-decay form:

$$D' = U + U \cdot R_D^\star \left(1 - e^{-(E-1)/R_D^\star}\right).$$

here $R_D^*$ represents the half-life of data reuse, repeating data beyond $R_D^*$ epochs will result in significant diminishing returns. This form approximates the geometric sum well and captures diminishing returns over repeated epochs. As the number of epochs $E \to \infty$, the exponential term vanishes and $D'$ asymptotically approaches: $D' \to U + U \cdot R_D^\star$, implying that no matter how many times data is repeated, the maximum usable signal is bounded by $(1 + R_D^\star) \cdot U$. This defines a natural saturation point on returns: even infinite compute yields no additional effective data beyond this limit.

A symmetric formulation is applied to model parameters for mathematical convenience which is used to define $N'$. Finally, a modified Chinchilla-style loss function incorporates these effective quantities $N'$ and $D'$:

$$\mathcal{L}(N, D) = \frac{A}{(N')^\alpha} + \frac{B}{(D')^\beta} + E_0,$$

with $A, B, \alpha, \beta, E_0, R_D^\star, N_D^\star$ fitted empirically from training runs. This formulation accurately captures loss behavior in regimes where data is reused multiple times and serves as a powerful tool for guiding training under data scarcity.

In this work, we adopt this framework to study how diffusion models and autoregressive models compare in their ability to extract value from repeated data, enabling apples-to-apples comparisons across compute, data, and model scale.

## 2.4 Training setup

We use the English C4 corpus [29], tokenized with the GPT-2 BPE vocabulary and truncated or padded to 2048 tokens per sequence. We consider unique-token budgets of $U \in \{25, 50, 100\}$M and train for up to 800 epochs (80B tokens total). Models are trained ranging from 7M to 2.5B parameters, following the Chinchilla scaling strategy where both width and depth are increased proportionally. The detailed architectural configurations of each model are provided in Appendix 9. For all training runs, we adopt the hyperparameter configuration introduced by Muennighoff *et al.* [21]. This may provide a slight advantage to autoregressive models, as these hyperparameters were originally tuned for that family. For details on hyperparameters please refer to Section 8 in Appendix.

## 3  Experiments

Our goal is to compare the performance of masked diffusion models and autoregressive models in data-constrained settings. To this end, we train a total of 200 models—100 diffusion models and 100 autoregressive models—across varying unique data sizes, model scales, and epoch counts. We present the empirical results in Section 3.1. In Section 3.2, we fit scaling laws tailored to data-constrained regimes for both model types, following the methodology introduced by Muennighoff *et al.*[21]. These scaling laws allow us to analyze performance trends and identify scenarios where diffusion models should be preferred over autoregressive ones (Section 3.3). In Section 3.4, we demonstrate that the superior validation loss of diffusion models indeed correlates with improved downstream task performance. Finally, in Section 3.5 we investigate the underlying cause of diffusion's advantage in data-constrained settings, showing that its exposure to diverse token orderings enables better generalization than AR's fixed left-to-right factorization.

### 3.1  Does Diffusion Beat AR in Data-Constrained Settings?

Prior comparisons between diffusion and autoregressive (AR) language models have largely focused on the single-epoch regime, where each token is seen only once during training [22, 39]. In this setting, diffusion models are consistently reported to require substantially more training compute ($C \sim 6ND$) than AR models to achieve comparable validation loss. For instance, Nie *et al.* [22] and Swerdlow *et al.* [39] derive scaling laws showing that masked diffusion models can require up to $16\times$ more compute than AR counterparts.

Crucially, these studies scale compute by increasing both the model size ($N$) and the amount of unique training data ($D$) proportionally. As a result, they do not isolate whether diffusion's 16x inefficiency stems from needing more total compute—or more unique data.

In other words: is diffusion limited by compute efficiency or by sample efficiency?

To answer this, we systematically study diffusion models in data-constrained settings, where the total amount of unique data is fixed and models are trained for many epochs, reusing the same data. Unlike prior work, our evaluation explicitly decouples model scaling from data reuse, allowing us to disentangle the effects of compute and data.

In Figure 1, we report empirical validation loss as a function of training FLOPs for the 50M and 100M regimes; results for the 25M setting are shown in Appendix Figure 9. We find that AR models initially outperform diffusion models when trained with the compute-optimal budget prescribed by Chinchilla scaling laws (denoted by the solid vertical line). However, this advantage disappears as training continues beyond this point. When models are allowed to train for additional epochs on repeated data, diffusion models consistently surpass AR models in validation loss across all data regimes. These findings indicate that the previously observed inefficiency of diffusion models is largely a consequence of evaluating them solely in the single-epoch regime. In data-constrained settings with repeated exposures, diffusion models extract significantly more value from the same data than their AR counterparts.

A key question remains is how should one go about increasing compute for diffusion models: by increasing model size, or by increasing the number of epochs (i.e., data reuse)? To address this, we analyze the trade-off between parameters and epochs in Figure 2, which shows validation loss contours as a function of both axes. In the 100M unique token regime, for example, we find that diffusion achieves its best loss at 500 epochs, while AR model reach its best at just 50 epochs. Each

point on the contour plot corresponds to a model trained with a specific parameter count and number of epochs; we report the actual validation loss at each configuration, without early stopping. We find that autoregressive models begin to overfit at high epoch counts, with validation loss worsening as training continues beyond a certain point. In contrast, diffusion models show no signs of overfitting within our compute budget—the best validation loss is achieved at the highest epoch counts we explore. This suggests that diffusion models continue to benefit from additional training on repeated data, and that observing overfitting may require significantly more compute.

To contextualize these results, we highlight two key configurations in Figure 2 for each model family: the compute-optimal point for single-epoch training, as identified by prior scaling law analyses [12, 24] (marked with a colored star in the bottom-left), and the best validation loss achieved under extended multi-epoch training (marked with a black star). At the compute-optimal point, which corresponds to training for a single epoch, diffusion models perform substantially worse than autoregressive models (10.65 vs. 7.07), consistent with prior findings that diffusion performs worse initially. However, as training is extended to hundreds of epochs, diffusion models continue to improve and eventually achieve a lower validation loss (3.55) than the best AR models (3.71).

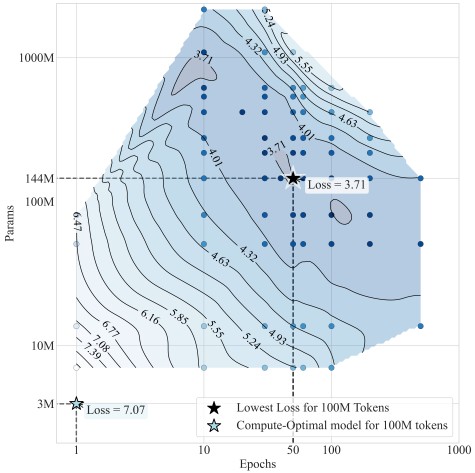

(a) Autoregressive contour: validation loss over epochs and model sizes.

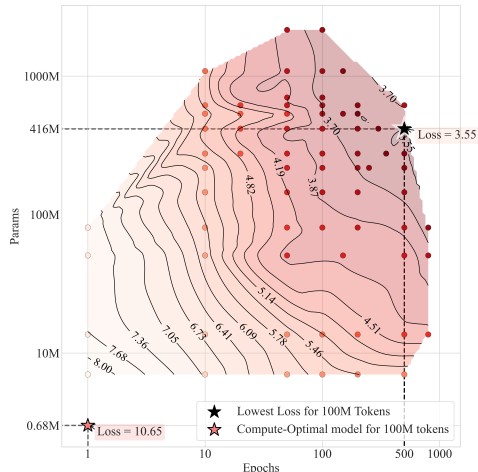

(b) Diffusion contour: validation loss over epochs and model sizes.

Figure 2: Validation loss contours over epochs and model sizes for autoregressive (left) and diffusion (right) models, trained on 100M unique tokens. Each plot shows validation loss as a function of training epochs (x-axis) and model parameters (y-axis). The colored star marks the compute-optimal point for single-epoch training, as predicted by prior scaling laws [12, 24], and the black star indicates the lowest validation loss achieved through extended multi-epoch training. In the single-epoch regime, diffusion models perform worse than AR models (10.65 vs. 7.07). However, when trained longer, diffusion models achieve a substantially lower final loss (3.55 vs. 3.71). This corresponds to a 67% reduction in loss for diffusion models compared to just 48% for AR models, highlighting their superior ability to leverage repeated data.

## 3.2 Fitting Data-Constrained Scaling Laws

To gain deeper insight into the trade-offs between diffusion and autoregressive models in data-constrained settings, we fit scaling laws to both model families across single-epoch and multi-epoch regimes, as described in Section 2.3. Our approach systematically varies three key factors: (1) the amount of unique data, (2) model parameter count, and (3) number of training epochs. This grid search allows us to disentangle the effects of data quantity, model capacity, and data reuse on final model performance.

We evaluate the quality of our scaling law fits using the coefficient of determination ($R^2$) and relative prediction error, as shown in Table 1. For autoregressive models, our $R^2$ values closely match those reported by Muennighoff *et al.* [21], indicating consistent behavior under repeated training. Interestingly, diffusion models yield significantly higher $R^2$ values, reflecting a better overall fit. We

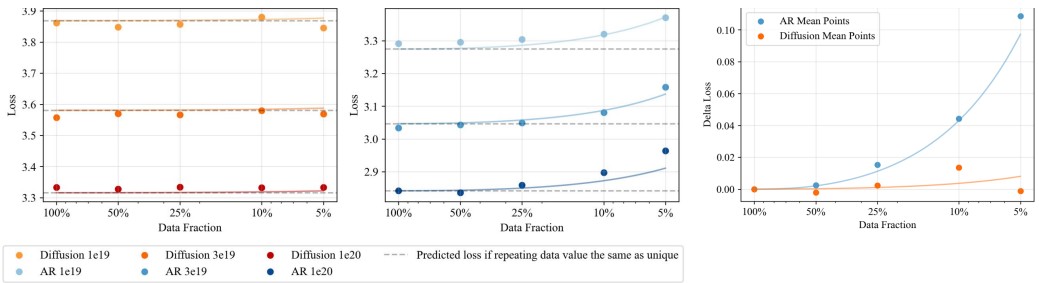

Figure 3: Decay rate of data value under repetition: left shows diffusion, middle AR, and right the average decay rate for both. Points are empirical results (darker color = higher FLOPs, lighter color = lower FLOPs; each line = fixed compute), we find that fitted curves (represented as lines) closely match the empirical points, indicating our scaling laws are representative. The decay rate of value for repeated data is lower for diffusion, reflecting its greater robustness to repeating.

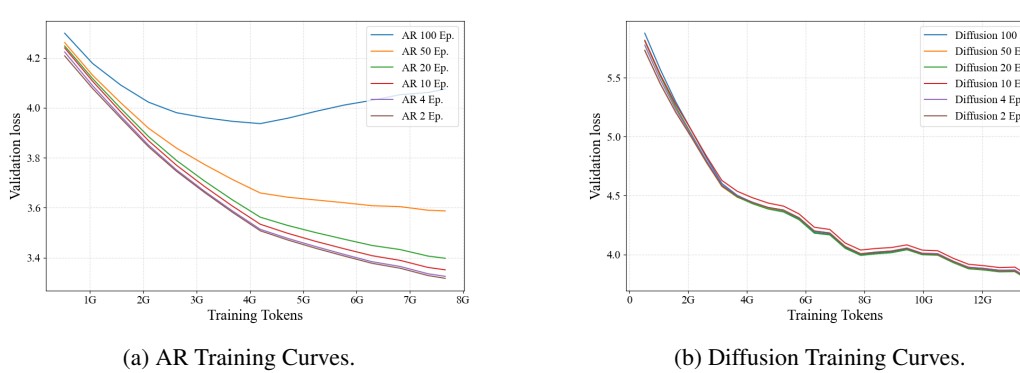

(a) AR Training Curves.  (b) Diffusion Training Curves.

Figure 4: Training curves for different epoch counts, all with using the same total compute. Each curve shows a different tradeoff between unique data and repetition. For AR models, validation loss rises with more epochs (overfitting), while for diffusion models, the curves are nearly unchanged, showing much greater robustness to data repetition.

Table 1: Fitting metrics of the scaling law model for Diffusion and AR. Diffusion and AR achieve a strong fit across both phases.

(a) Initial fit.

| Model | $R^2$ | Loss |
|-------|-------|------|
| Diffusion | 0.9447 | 0.0002 |
| AR | 0.9439 | 7.7532e−05 |

(b) Second step fit with extracted scaling parameters.

| Model | $R^2$ | Loss | $R_D^*$ | $R_N^*$ |
|-------|-------|------|---------|---------|
| Diffusion | 0.9784 | 0.00079 | 493.89 | 1265.65 |
| AR | 0.7628 | 0.00361 | 31.19 | 55.16 |

attribute this to lower variance in validation loss across training runs, likely due to the absence of overfitting in diffusion models even at high epoch counts.

Beyond the overall fit, we extract two key parameters from the scaling laws: $R_D^*$, which characterizes the effective half-life of data reuse—i.e., the number of epochs after which additional training on repeated data yields diminishing returns—and $R_N^*$, which indicates the optimal model size for a given data budget. Our results reveal a sharp contrast in data reuse half-lives: diffusion models exhibit an $R_D^*$ of 512.85, compared to just 31.93 for autoregressive models. A higher $R_D^*$ implies that a model can benefit from many more repeated exposures before saturating. This suggests that diffusion models continue to improve across hundreds of epochs, while AR models quickly saturate—highlighting the superior sample efficiency of diffusion models in data-constrained regimes.

Figure 3 illustrates how the utility of unique data decays with increased repetition. We evaluate this effect across three compute budgets—$1 \times 10^{19}$, $3 \times 10^{19}$, and $1 \times 10^{20}$ FLOPs—by varying the proportion of unique data and parameters while keeping total compute fixed (e.g., 50% of the data

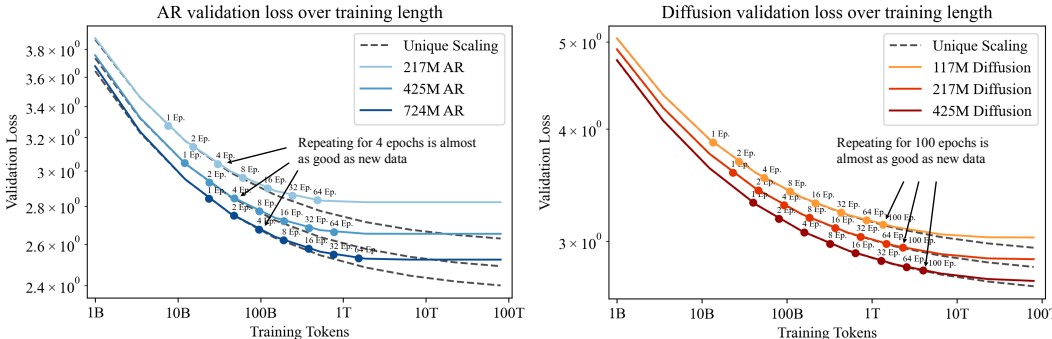

Figure 5: Predicted validation loss for AR (left) and Diffusion models (right) under compute-optimal settings, extrapolated to larger compute budgets. Dotted lines show the hypothetical case where repeated data equals new data. For AR, this holds up to ≈4 epochs; for diffusion, up to ≈100 epochs—showing diffusion's greater robustness to data repetition. Note that loss values between AR and diffusion are not directly comparable, as they're extrapolated from scaling laws with different data-entropy terms ($E_0$). In Section 3.5, we ignore this factor during comparison.

for 2 epochs, 25% for 4 epochs, etc.). For each compute budget, we use single-epoch scaling laws to determine the optimal model size and unique token count for both AR and diffusion models. We present both empirical results and fitted curves from our parametric scaling law, observing strong agreement between the two. Notably, the decay rate of data value remains consistent across compute budgets. Diffusion models consistently exhibit a substantially slower decay rate than AR models, suggesting they are better able to extract value from repeated data.

Figure 4 shows validation loss versus training tokens using the compute budget of 1e19. The results reinforces the trend: AR models overfit with increased repetition, showing diverging loss curves. In contrast, diffusion models exhibit overlapping curves across repetitions, indicating no signs of overfitting and a very low decay rate with data reuse.

Figure 5 shows extrapolated training curves at large compute budgets. For each setting, we use the compute-optimal model and dataset size derived from single-epoch scaling laws for 1e19, 3e19 and 1e20. We then extend training to multiple epochs. The dashed lines represent the ideal Chinchilla-style scaling behavior, where all training tokens are assumed to be unique. We find that for AR models, repeated data provides nearly the same benefit as fresh data only up to about 4 epochs. Beyond this point, additional repetition yields diminishing returns. In contrast, diffusion models continue to match the unique-data curve for up to 100 epochs, indicating a far greater capacity to benefit from repeated data in data-constrained regimes.

## 3.3   When to Use Diffusion over AR?

A key question for practitioners is: *when should diffusion be preferred over autoregressive models (AR)?* To answer this, we compare the fitted data-constrained scaling laws for both model families (§2.3).

We define the validation loss gap between diffusion and AR as:

$$\Delta\mathcal{L}(C, U) = \mathcal{L}_{\text{Diffusion}}(C, U) - \mathcal{L}_{\text{AR}}(C, U),$$

where $C$ is total training compute and $U$ is the number of unique tokens. Positive values favor AR; negative values favor diffusion. The *critical compute* $C_{\text{crit}}(U)$ is the point where the models perform equally: $\Delta\mathcal{L}(C_{\text{crit}}, U) = 0$.

Figure 6(a) shows a heatmap of $\Delta\mathcal{L}$ over compute and data. Red regions indicate regimes where diffusion outperforms AR ($\Delta\mathcal{L} < 0$), while blue regions favor AR. As expected, AR performs better in low-compute settings due to its efficient per-step learning. However, diffusion models begin to outperform AR at higher compute, especially when data is limited and repeated.

Figure 6(b) plots the **critical compute frontier** $C_{\text{crit}}(U)$—the compute required for diffusion to match AR at a given unique token count $U$. This frontier follows a power law: $C_{\text{crit}}(U) \propto U^{2.174}$.

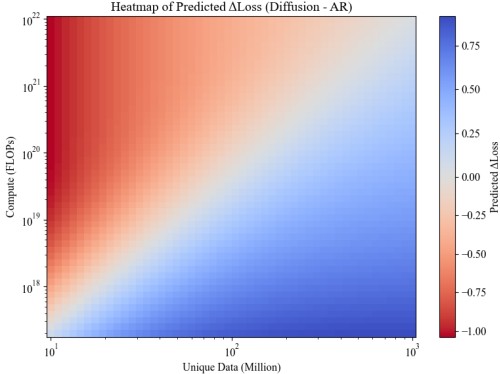
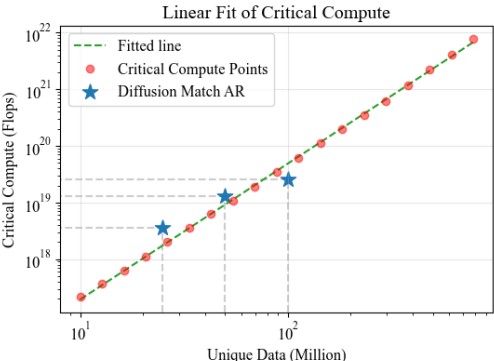

(a) **Loss Gap Heatmap.** Difference in validation loss ($\Delta\mathcal{L} = \mathcal{L}_{\text{Diffusion}} - \mathcal{L}_{\text{AR}}$) across unique data sizes and FLOPs. Red indicates regions where diffusion outperforms AR models and blue where AR outperforms diffusion.

(b) **Critical Compute Curve.** The FLOPs threshold $C_{\text{crit}}(U)$ beyond which diffusion outperforms AR models. This follows a power law: $C_{\text{crit}}(U) \propto U^{2.174}$.

Figure 6: **When does Diffusion beat AR?** Left: Heatmap showing where diffusion models have lower validation loss than AR models. Right: The critical compute curve defining the compute threshold needed for diffusion to match autoregressive models at a given unique token count.

The linear fit in log-log space is:

$$\log_{10}(U) = 0.460 \cdot \log_{10}(C) - 1.050, \quad \text{so} \quad C_{\text{crit}}(U) = 2.12 \times 10^{1.956} \cdot U^{2.174}.$$

The dark green dashed line shows the fitted curve, and the blue stars represent empirical crossover points—where diffusion matches AR performance in experiments. These points align closely with the predicted frontier, confirming our fitted equation's accuracy.

## 3.4 Downstream Results

| Benchmarks | Random Baseline | AR | AR (Flop matched) | Diffusion |
|---|---|---|---|---|
| ARC-Easy [6] | 25.00 | 35.63 | 35.35 | **37.84** |
| BoolQ [5] | 50.00 | 46.00 | 38.23 | **49.38** |
| COPA [30] | 50.00 | 56.33 | 54.00 | **59.00** |
| HellaSwag [45] | 25.00 | 27.37 | 29.03 | **30.24** |
| PiQA | 50.00 | 60.94 | **61.64** | 60.72 |
| RACE [16] | 25.00 | 25.28 | 24.88 | **28.96** |
| WinoGrande XL [32] | 50.00 | 48.87 | 49.41 | **50.97** |
| SciQ [14] | 25.00 | 58.05 | 50.50 | **68.67** |
| Lambada [27] | 00.00 | 10.91 | 5.53 | **15.19** |

*Note:* All values represent accuracy (%). Best results shown in bold.

Table 2: Downstream results for the best-performing (as per validation loss) and flop-matched autoregressive (AR) and diffusion models trained with 100M unique tokens. Random baselines are reported for reference.

We evaluate the best-performing diffusion and autoregressive (AR) models, selected based on their validation loss, across several downstream benchmarks to examine whether lower validation loss translates to improved generalization.

Since AR models tend to overfit much earlier, we additionally evaluate *flop-matched overfitted* AR models trained for the same number of epochs as their diffusion counterparts. On a few set of benchmarks, these overfitted AR models slightly outperform their best-validation variants; however, across almost all tasks, diffusion models consistently achieve the highest downstream performance.

Additional results, with 500M unique tokens and other downstream datasets, are provided in Table 3 and Table 4 in the Appendix.

Across a diverse set of tasks and data scales, diffusion models consistently outperform their AR counterparts.

### 3.5 Why do Diffusion models outperform AR models in data-constrained settings?

To better understand why diffusion models are more sample-efficient than autoregressive (AR) models, we conducted a series of controlled experiments aimed at isolating the core source of diffusion's advantage.

We first applied standard perturbation-based techniques during AR training. Specifically, we used: (i) attention dropout — randomly dropping 25%, 50%, or 75% of attention weights; and (ii) token masking — masking a subset of input tokens by zeroing their attention weights across all layers, while retaining the standard next-token prediction objective.

As shown in Figures 8a and 8b in Appendix, neither approach improved validation loss. In all cases, AR models continued to overfit and remained far behind diffusion models trained for longer epochs. All AR baselines here used 140M parameters and were trained for 50 epochs; the red line in the plots marks the best diffusion model from Figure 2b, trained for 500 epochs.

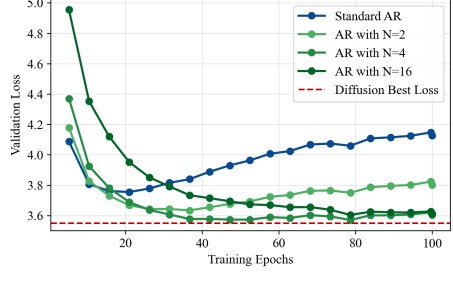

Figure 7: Validation loss improves as the number of token orderings $N$ increases in AR training. At $N = 16$, performance approaches that of diffusion models.

We next investigated whether diffusion's advantage stems from exposure to diverse token orderings. To test this, we trained AR models with varying numbers of orderings: $N = 1$ denotes standard left-to-right training, while $N = k$ adds $k-1$ random permutations of the sequence order. All permutations were fixed prior to training, and each training batch randomly samples from these orderings. All AR models in this setting used 278M parameters and were trained for 100 epochs. As shown in Figure 7, increasing $N$ consistently lowered validation loss and delayed overfitting. At $N = 16$, the 100-epoch validation loss of AR models approached that of diffusion, suggesting that diverse orderings are indeed a key driver of diffusion's sample efficiency.

These results support our interpretation that diffusion models outperform AR models in low-data regimes because they are implicitly trained on a richer distribution of conditional prediction tasks.

Finally, this analysis suggests a natural continuum between the two paradigms: by controlling task diversity—through masking or reordering—we could design hybrid models that interpolate between compute efficiency (AR-like) and sample efficiency (diffusion-like). Exploring this continuum is a promising direction for future work. Details of our permutation process are in Section 12.

## 4 Conclusion

As the availability of high-quality data plateaus, improving sample efficiency becomes essential for scaling deep learning. In this work, we show that masked diffusion models consistently outperform autoregressive (AR) models in data-constrained regimes — when training involves repeated passes over a limited dataset. We establish new scaling laws for diffusion models, revealing their ability to extract value from repeated data far beyond what AR models can achieve. These results challenge the conventional belief that AR models are universally superior and highlight diffusion models as a compelling alternative when data—not compute—is the primary bottleneck. Looking ahead, efficient use of finite data may define the next frontier in scaling deep learning models. Although the studies have been performed in the context of language models, we believe these findings should apply across any kind of sequence modeling data, such as in robotics or healthcare.

For practitioners, our takeaway is simple: ***if you are compute-constrained, use autoregressive models; if you are data-constrained, use diffusion models.***

## 5 Acknowledgement

We thank Alexander Li for his valuable insights and detailed feedback on the manuscript. We are also grateful to Zheyang Qin for his help in improving the figures. Finally, we thank Niklas Muennighoff, Simo Ryu and Colin Raffel for their helpful comments on the final draft of the paper. We thank Stella Biderman for pointing out the inconsistent unit used in the Critical Compute equation. We thank Lucas Beyer and You Jiacheng, in suggesting the experiments in Section 3.5, that helped us explain why diffusion models would be more sample-efficient. This work was supported in part by ONR MURI N00014-22-1-2773, ONR N00014-22-1-2096, and AFOSR FA9550-23-1-0747. The results and models presented in this work also used compute resources from the National AI Research Resource Pilot, with support from NVIDIA, including NVIDIA's DGX Cloud product and the NVIDIA AI Enterprise Software Platform.

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

# Appendix

## 6  Related Work

**Deep Learning in Data-Constrained Settings.**  Deep learning progress has been largely driven by the scaling of both data and compute. However, recent analyses suggest we may soon face a data bottleneck that could inhibit continued advancement [42]. In language modeling, the dominant paradigm has been autoregressive (AR) models [41, 28, 4], which are typically trained for a single epoch to maximize exposure to unique tokens [11]. In light of looming data constraints, Muennighoff et al.[21] show that AR models can still benefit from data reuse: training for up to four epochs on repeated data achieves performance nearly on par with training on fresh data, suggesting an effective strategy for improving data efficiency. In contrast, computer vision has long embraced multi-epoch training along with aggressive data augmentation—such as random cropping, flipping, and color jittering—to expand effective dataset size and improve generalization[36, 43], particularly for discriminative tasks like classification and detection. Despite these practices, data efficiency in generative modeling remains underexplored, and the trade-offs between leading paradigms such as diffusion and AR models under constrained data regimes are still poorly understood.

**Diffusion-Based Language Models.**  Diffusion models, originally developed for image generation [10], have recently been adapted to text, offering a fundamentally different paradigm for language modeling [2, 17, 9]. Broadly, diffusion language models fall into two categories: *continuous* and *discrete*. Continuous approaches [9] inject Gaussian noise in the forward process, whereas discrete methods [2] corrupt tokens with noise sampled from distributions such as Bernoulli. Among the two classes, continuous diffusion has proven more difficult to scale on language data [9, 19]. In contrast, recent advances in *discrete* diffusion—particularly masked diffusion—have shown encouraging results. Recent work [1, 7, 31, 19] has significantly narrowed the performance gap between diffusion and AR models. Notably, LLaDA [23] scales masked diffusion models to 8B parameters and achieves results similar to LLaMA3-8B across both pretraining and instruction-tuned evaluations. Furthermore, Nie et al. [22] provide scaling law analysis showing that diffusion models follow similar power-law trends as AR models, though they may require up to $16\times$ more compute under single-epoch training, Swerdlow et al. [39] find similar trends on multimodal data containing both image and text. However, these evaluations are restricted to single-pass training and do not examine the data-constrained, multi-epoch regimes which is the focus of our work.

## 7  Additional Results

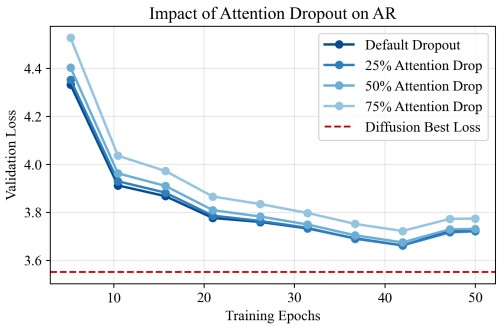

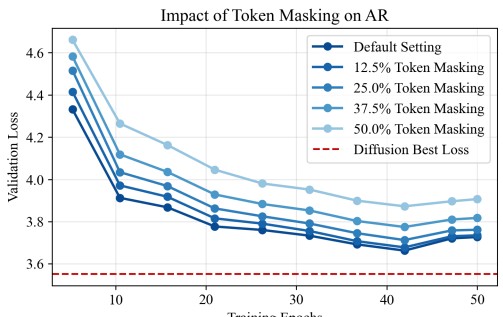

(a) Validation loss under varying attention dropout levels in AR training.

(b) Validation loss under varying token masking levels in AR training.

Figure 8: Impact of common data augmentations on AR models. Despite applying attention dropout and token masking, AR models still overfit and underperform compared to diffusion models. We believe this gap arises because diffusion models learn random factorizations of the joint distribution, rather than a fixed left-to-right ordering.

In Table 3, we report downstream results using 500M unique tokens. Guided by the critical compute threshold derived in Section 3.3, we scale training to 500M unique tokens and train a 2.3B-parameter

diffusion model under the predicted compute budget. The model is trained for 130 epochs, after which we observe no signs of convergence and terminate due to compute limitations. For the autoregressive (AR) model, we report results from the checkpoint that achieves the best validation loss before overfitting..

Figure 8a and Figure 8b show that standard perturbation-based techniques, such as attention dropout (25–75%) and token masking, fail to improve validation loss. Across all settings, AR models continue to overfit and lag behind diffusion models trained for longer epochs. All AR baselines use 140M parameters and are trained for 50 epochs, whereas the red line denotes the best diffusion model from Figure 2b, trained for 500 epochs.

Table 4 reports the negative log-likelihood (NLL; lower is better) on four diverse corpora: OSCAR [25], TinyStories[8], WikiText [20], and IndustryCorpus2 EN Sub [35]. These datasets span open-domain, narrative, encyclopedic, and industry-specific text.

Figure 9 shows validation loss versus training FLOPs for autoregressive (AR) and masked diffusion models under data-constrained settings. This figure extends Figure 1 in the main paper, by including empirical validation loss for the 25M unique token regimes.

Table 3: Downstream results for the best-performing (as per validation loss) autoregressive and diffusion models trained with 500M unique tokens. We also report the random baseline for reference.

| Benchmarks | Random Baseline | AR | Diffusion |
|---|---|---|---|
| ARC-Easy [6] | 25.00 | 43.79 | **45.95** |
| BoolQ [5] | 50.00 | 51.87 | **55.26** |
| COPA [30] | 50.00 | **67.00** | 64.83 |
| HellaSwag [45] | 25.00 | 32.28 | **35.33** |
| PiQA | 50.00 | **65.71** | 65.61 |
| RACE [16] | 25.00 | 28.28 | **31.44** |
| WinoGrande XL [32] | 50.00 | 50.61 | **51.51** |
| SciQ [14] | 25.00 | 67.82 | **79.13** |
| Lambada [27] | 00.00 | 15.07 | **22.30** |

*Note:* All values represent accuracy (%). Best results shown in bold.

Table 4: Downstream NLL of best diffusion and AR models at 100M unique data points.

| Model Type | Flops | OSCAR | TinyStories | WikiText | IndustryCorpus2 |
|---|---|---|---|---|---|
| Best ARM | 4.32e18 | 3.98 | 2.96 | 4.94 / 4.96 | 3.58 |
| Best MDM | 1.24e20 | **3.83** | **2.93** | **4.50 / 4.52** | **3.44** |

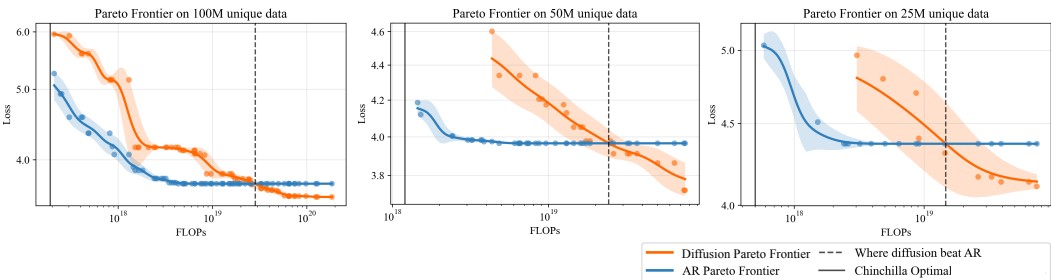

Figure 9: Pareto frontier of validation loss (negative log-likelihood) versus training FLOPs for autoregressive (AR) and diffusion models under data-constrained settings, on three different unique data settings 25M, 50M and 100M.

# 8   Hyperparameter details

We use the following hyperparameters: batch size of 256 sequences, AdamW optimizer with $\beta_1$=0.9, $\beta_2$=0.95, $\epsilon$=$10^{-8}$, a learning rate schedule with peak 2e-4, minimum 2e-5, 1% warm-up, cosine decay, weight decay 0.1, and gradient clipping of 1.0.

# 9   Model Architecture

We adopt the Megatron-DeepSpeed framework as the foundation of our implementation, upon which we build our training and evaluation setup for the masked Diffusion Model. Similar to the "extended version of the architectures" proposed in [22], our model adheres to the general transformer design while introducing several architectural modifications to better align with modern LLM practices.

Specifically, we replace absolute positional embeddings with Rotary Positional Embeddings (RoPE) [37], which improve extrapolation to longer contexts and reduce parameter count. Furthermore, we adopt the SwiGLU activation function in the MLP blocks, which has been shown to outperform standard GELU or ReLU in both convergence and downstream performance [33]. To further simplify the architecture and enhance training stability, we substitute standard LayerNorm with RMSNorm and eliminate all bias terms. These design choices are consistent with [3, 40].

To preserve the original MLP capacity while aligning with hardware-friendly parameter sizes, we compute the feed-forward hidden size $h_f$ as:

$$h_f = \left\lfloor \frac{8 \cdot d_{\text{model}}}{3 \cdot 64} \right\rfloor \cdot 64$$

This rounding scheme ensures that the FFN hidden size remains divisible by 64 while closely matching the effective dimensionality used in SwiGLU layers.

We slightly modify the parameter count estimation formula from the original:

$$P = 12lh^2 \left( 1 + \frac{13}{12h} + \frac{V+s}{12lh} \right)$$

to better reflect our revised architecture. The original formula can be decomposed into: $4lh^2$ (attention), $8lh^2$ (MLP), $13lh$ (LayerNorm and biases), and $(V+s)h$ (token and positional embeddings). After applying our architectural adjustments—namely, using a SwiGLU-based MLP of dimension $h_f$, switching to RoPE (eliminating $sh$), and removing bias terms—we arrive at the revised formula:

$$P = 4lh^2 + 3lh \cdot h_f + 6lh + Vh$$

Table 5 presents all model configurations used in our experiments along with their parameter counts.

# 10   Discussion

**Why are autoregressive (AR) models more compute-efficient than diffusion models?**   We hypothesize two main contributing factors. (i) Order specialization: AR models are trained with a fixed left-to-right factorization, so every gradient update reinforces the same prediction task, allowing them to specialize effectively. In contrast, diffusion models must generalize across many random token orderings, which hinders specialization. (ii) Stronger supervision per update: In AR training, every token in a training sequence serves as a supervised target, and the causal structure enables dense gradient updates, resulting in stable, low-variance learning. Diffusion models, however, compute loss only on a subset of masked tokens, making supervision sparser per sequence, even though gradients propagate through the entire input. As a result, each update carries less direct learning signal. Arriola et al. [1] show that tuning the masking schedule can help reduce gradient variance and improve training compute efficiency.

Table 5: Model Architectures

| Name | param (M) | d_model | origin_ffw_size | ffw_size | kv_size | n_heads | n_layers |
|---|---|---|---|---|---|---|---|
| 7 | 7.0 | 128 | 512 | 320 | 32 | 4 | 3 |
| 14 | 13.6 | 224 | 896 | 576 | 32 | 7 | 4 |
| 20 | 19.5 | 288 | 1152 | 768 | 32 | 7 | 5 |
| 35 | 36.6 | 448 | 1792 | 1152 | 32 | 7 | 6 |
| 44 | 50.7 | 512 | 2048 | 1344 | 64 | 8 | 8 |
| 57 | 64.8 | 576 | 2304 | 1536 | 64 | 9 | 9 |
| 74 | 80.5 | 640 | 2560 | 1664 | 64 | 10 | 10 |
| 90 | 95.0 | 640 | 2560 | 1664 | 64 | 10 | 13 |
| 106 | 109.6 | 640 | 2560 | 1664 | 64 | 10 | 16 |
| 117 | 123.6 | 768 | 3072 | 2048 | 64 | 12 | 12 |
| 140 | 144.8 | 768 | 3072 | 2048 | 64 | 12 | 15 |
| 163 | 166.1 | 768 | 3072 | 2048 | 64 | 12 | 18 |
| 175 | 179.2 | 896 | 3584 | 2368 | 64 | 14 | 14 |
| 196 | 198.3 | 896 | 3584 | 2368 | 64 | 14 | 16 |
| 217 | 217.5 | 896 | 3584 | 2368 | 64 | 14 | 18 |
| 251 | 250.8 | 1024 | 4096 | 2688 | 64 | 16 | 16 |
| 278 | 275.7 | 1024 | 4096 | 2688 | 64 | 16 | 18 |
| 306 | 300.6 | 1024 | 4096 | 2688 | 64 | 16 | 20 |
| 425 | 416.9 | 1280 | 5120 | 3392 | 128 | 10 | 18 |
| 489 | 475.6 | 1280 | 5120 | 3392 | 128 | 10 | 21 |
| 509 | 495.9 | 1408 | 5632 | 3712 | 128 | 11 | 18 |
| 552 | 534.4 | 1280 | 5120 | 3392 | 128 | 10 | 24 |
| 587 | 566.7 | 1408 | 5632 | 3712 | 128 | 11 | 21 |
| 632 | 615.3 | 1536 | 6144 | 4096 | 128 | 12 | 19 |
| 664 | 637.6 | 1408 | 5632 | 3712 | 128 | 11 | 24 |
| 724 | 700.3 | 1536 | 6144 | 4096 | 128 | 12 | 22 |
| 816 | 785.2 | 1536 | 6144 | 4096 | 128 | 12 | 25 |
| 893 | 856.4 | 1792 | 7168 | 4736 | 128 | 14 | 20 |
| 1018 | 971.3 | 1792 | 7168 | 4736 | 128 | 14 | 23 |
| 1143 | 1086.3 | 1792 | 7168 | 4736 | 128 | 14 | 26 |
| 1266 | 1207.6 | 2048 | 8192 | 5440 | 128 | 16 | 22 |
| 1424 | 1353.6 | 2176 | 8704 | 5760 | 128 | 17 | 22 |
| 1429 | 1358.2 | 2048 | 8192 | 5440 | 128 | 16 | 25 |
| 1593 | 1508.9 | 2048 | 8192 | 5440 | 128 | 16 | 28 |
| 1609 | 1523.2 | 2176 | 8704 | 5760 | 128 | 17 | 25 |
| 1731 | 1644.9 | 2304 | 9216 | 6144 | 128 | 18 | 24 |
| 1794 | 1692.9 | 2176 | 8704 | 5760 | 128 | 17 | 28 |
| 2007 | 1899.8 | 2304 | 9216 | 6144 | 128 | 18 | 28 |
| 2283 | 2154.7 | 2304 | 9216 | 6144 | 128 | 18 | 32 |
| 2298 | 2165.3 | 2560 | 10240 | 6784 | 128 | 20 | 26 |
| 2639 | 2478.6 | 2560 | 10240 | 6784 | 128 | 20 | 30 |
| 2980 | 2791.9 | 2560 | 10240 | 6784 | 128 | 20 | 34 |
| 3530 | 3257.0 | 2688 | 10752 | 7168 | 128 | 21 | 36 |
| 3802 | 3561.3 | 2816 | 11264 | 7488 | 128 | 22 | 36 |
| 4084 | 3879.2 | 2944 | 11776 | 7808 | 128 | 23 | 36 |
| 4516 | 4231.9 | 3072 | 12288 | 8192 | 128 | 24 | 36 |
| 6796 | 6337.4 | 3584 | 14336 | 9536 | 128 | 28 | 40 |
| 9293 | 8640.6 | 4096 | 16384 | 10880 | 128 | 32 | 42 |
| 11452 | 10889.0 | 4352 | 17408 | 11584 | 128 | 32 | 47 |
| 12295 | 11444.2 | 4608 | 18432 | 12288 | 128 | 36 | 44 |
| 12569 | 12208.7 | 4608 | 18432 | 12288 | 128 | 32 | 47 |
| 13735 | 13560.0 | 4864 | 19456 | 12928 | 128 | 32 | 47 |
| 14940 | 14905.3 | 4992 | 19968 | 13312 | 128 | 32 | 49 |
| 16183 | 15028.3 | 5120 | 20480 | 13632 | 128 | 40 | 47 |

---
**Algorithm 1** Generating a Random Order List with Predefined Permutations
---
**Input:** Sequence length $L$, number of orders $N$, random seed $s$
**Output:** Order list $\mathcal{O}$ of $N$ orderings
---
 1: Initialize order list $\mathcal{O} \leftarrow []$
 2: Append raster order: $\mathcal{O} \leftarrow \mathcal{O} \cup \{[0, 1, \ldots, L-1]\}$
 3: **for** $i = 1$ to $N - 1$ **do**
 4:     $b \leftarrow [0, 1, \ldots, L-1]$ {base raster order}
 5:     $\epsilon \sim \mathcal{N}(0, i^2 I)$ {add Gaussian noise with scale $i$}
 6:     $s \leftarrow b + \epsilon$ {perturbed scores}
 7:     $\pi \leftarrow \text{argsort}(s)$ {permutation order}
 8:     $\mathcal{O} \leftarrow \mathcal{O} \cup \{\pi\}$
 9: **end for**
10:
11: **return** $\mathcal{O}$
---

## 11 Limitations

In this work, we examined two extremes of generative modeling: masked diffusion models, which learn over random condition prediction tasks and are more data-efficient, and autoregressive (AR) models, which follow a fixed left-to-right order and are more compute-efficient. While our results highlight a clear trade-off, this need not be binary—hybrid models that interpolate between AR and diffusion would offer a better balance. Although prior works have explored such hybrids [1, 13], they have not been evaluated through the lens of data-compute efficiency. We explore part of this question in Section 3.5, however it will be useful to study this in more detail. Additionally, our scaling laws are currently fit over a limited range of unique data sizes; extending them to larger regimes may improve predictive accuracy and reveal further insights.

## 12 Order Permutation Details

In this experiment, we train autoregressive models using different token orderings. We do not introduce target positional embeddings as done in works such as RAR [44, 26]. We evaluated the trained models using left-to-right ordering. We define the perturbations in the token ordering by adding varying levels of noise to the left-to-right ordering.

Specifically, we generate a list of N orderings, where the first order is the standard left-to-right (l2r) order. Subsequent permutations are created by adding Gaussian noise to the left-to-right position ids, with the standard deviation of the noise directly proportional to the permutation's index. This method allows us to create a spectrum of orderings, from the standard l2r order to more heavily permuted sequences, as detailed in Algorithm 1.

During training, we apply these predefined orders to the input sequences. For each sequence in a batch, we randomly sample a permutation from our predefined list. This process is summarized in Algorithm 2 and further detailed below:

For each sequence, the first token is kept fixed. This ensures that the position ID 0 is always assigned to the first token, providing a soft absolute positional anchor for the sequence when using RoPE[38]. Under RoPE, attention depends only on relative position offsets rather than absolute information, i.e. $\langle R(i)q, R(j)k \rangle = qR(i-j)k$. Therefore, fixing position 0 on the first token keeps the control anchor unrotated $R(0) = I$ and removes global sequence-wise phase shifts induced by permutations, which stabilized the optimization and reduced variance under permutation augmentation.

As an example, suppose that the number of predicted tokens is $T$ (e.g. $T = 2048$ in our default setting) and the total input length is $L = T + 1$ including the label shift. Only the indices in $[1:T]$ are shuffled and assigned position IDs from $\{1, \ldots, T\}$. For instance, with $T = 6$ and a permutation $\pi = [2, 0, 1, 4, 5, 3]$, the resulting token and label orders are:

$$\text{tokens:} \quad [0, 3, 1, 2, 5, 6],$$
$$\text{labels:} \quad [3, 1, 2, 5, 6, 4].$$

**Algorithm 2** Shuffling Tokens Using Predefined Order Lists

---

**Input:** Token matrix $\texttt{tokens} \in \mathbb{Z}^{B \times L+1}$ (including last label), order list $\mathcal{O}$ of $K$ permutations
**Output:** Shuffled tokens and position IDs

1: Let $B \leftarrow$ number of sequences in batch
2: Let $L \leftarrow$ sequence length
3: Initialize $\texttt{position\_ids} \leftarrow \mathbf{0}^{B \times L}$
4: Sample index vector $I \sim \text{Uniform}(\{0, \ldots, K-1\})^B$ {select random order for each sequence}
5: **for** $i = 1$ to $B$ **do**
6:    $\pi \leftarrow \mathcal{O}[I_i]$ {retrieve $i$-th random order}
7:    $\texttt{tokens}[i, 1{:}] \leftarrow \texttt{tokens}[i, 1{:}][\pi]$ {shuffle tokens except first token}
8:    $\pi \leftarrow \pi + 1$ {shift positions by 1 to reserve position 0}
9:    $\texttt{position\_ids}[i, 1{:}] \leftarrow \pi[0{:}L{-}1]$ {assign shifted positions}
10: **end for**
11:
12: **return** $\texttt{tokens}, \texttt{position\_ids}$

---

# 13 Broader Impacts

Our findings suggest that masked diffusion models are more sample-efficient language modeling, which is especially valuable as high-quality textual data becomes scarce. This could benefit low-resource languages, privacy-sensitive domains, and scientific fields where data is limited or costly. By reducing dependence on massive proprietary datasets, diffusion models may help democratize access to LLM development. Nonetheless, the broader deployment of generative models raises concerns around misuse and misinformation, underscoring the need for responsible research and deployment practices.

