# OpenReview forum: "Diffusion Beats Autoregressive in Data-Constrained Settings"
_NeurIPS.cc/2025/Conference — NeurIPS 2025 poster_

### Official Review · Reviewer_5KGy · 2025-06-26

**Clarity:** 3
**Significance:** 3
**Originality:** 3
**Rating:** 5
**Confidence:** 4

**Summary:**

This paper examines scaling laws for masked diffusion models (MDMs) vs autoregressive models (ARMs) in data constrained regimes, where there are not enough unique tokens to train at “Chincilla” optimal scaling laws. Contrary to previous studies that have found MDMs lag behind ARMs in terms of compute efficiency, in data constrained regimes, MDMs can be trained for longer and attain superior language modeling performance compared to ARMs.

**Questions:**

1. Can the authors clarify Table 1? What is the difference between Table 1a and Table 1b? Why is the $R^2$ so poor in Table 1b?
2. In Figure 2, isn’t the perplexity (PPL) / NLL here relatively poor? For example, the reported best NLL corresponds to roughly 30 ppl. Other recent MDM papers are reporting even lower PPL for significantly smaller models (~150M param) on much larger datasets (OpenWebText); see for example Table 2 in MDLM [3]. Why is the final best PPL for the models in these experiments relatively poor?
3. I believe the authors are missing a scaling factor of $\frac{r(s) - r(t)}{1 - r(t)}$ for the MDM loss in line 127. Was this used during training / eval to make the loss a “proper” lower bound?
4. I don’t think I understand lines 128-130 regarding the additional pre-training loss to enable CFG. To the best of my understanding, the CFG from Nie et al. [6] was an inference-time only scheme. That is, the Nie et al. [6] used the same model trained with standard MDM objective and simply masked out the context/prompt to attain the unconditional distribution estimate. Please correct me if I am wrong here.
5. In Figure 2, is the colorful “Chinchila-optimal” star for 1 epoch of training?
6. Why are there different model sizes in the legends of Figure 5 for ARM vs MDMs?
7. In Section 5, line 310, can the authors clarify what is meant by “... showing masking, not attention, is key to their advantage when data is scarce”? Is this meant to say that the bi-directional attention of MDMs is not the key component? If this is the claim, then I would say it is not actually tested. You could validate this by training MDMs with causal attention and random masking?

[6] Nie, Shen, et al. "Scaling up Masked Diffusion Models on Text." arXiv preprint arXiv:2410.18514 (2024).

**Ethical Concerns:**

["NO or VERY MINOR ethics concerns only"]

**Final Justification:**

The authors clarified my concerns and provided extensive additional experimental results

**Limitations:**

The main limitation is detailed in the weaknesses above: the “data constrained” regime examined here seems to be potentially overly constrained / small and results may not extrapolate to larger unique token regimes.

**Quality:**

3

**Strengths And Weaknesses:**

### **Strengths**

This paper offers a unique lens on the tradeoffs between ARMs and MDMs, which are gaining in popularity. The experimental design is extensive and the results are overall reported in a useful manner that supports the authors’ main claims. Finding regimes where practitioners can tradeoff the advantages of MDMs vs ARMs is of high significance to the community.

### **Weaknesses**

1. I would have liked to see the discussion of underlying hypotheses that drive the results earlier on in the paper and play a more central part of the experimental design. At the moment, the one hypothesized driver is relegated to a few sentences in the discussion section. I even agree with the authors’ hypothesized explanation that MDMs serve as a form of “data augmentation” / “regularization”. I think this is the correct thinking, but I would have liked to see this hypothesis tested in some way through the experiments.
2. I find the motivation of running out of internet-scale data to be somewhat removed from the experiments conducted in the paper. The scales examined here of 100M tokens are orders of magnitude lower than the pre-training sizes of even the 8B parameter sized AR models (~30 trillions tokens, e.g. Qwen 3 [1] ) or diffusion models (e.g. Llada [2] at 2.3 trillion tokens).
3. On a related note, I am a bit concerned about how small the “data constrained” regimes are. For example, 25M unique tokens with 2048 sequence length is ~12k unique samples. This is a **very** small dataset. For example, even the widely used downstream task dataset of GSM8k has 8-9k samples in the training split.
4. I also believe that the authors should discuss the more recent Llada [2] work that showed a different perspective on scaling laws and rather than focusing on NLL/perplexity showed that for downstream tasks, MDMs have more comparable scaling laws to ARMs.
5. I disagree with the end of the first paragraph in Section 4.1 (lines 202-205). MDMs **are** more compute inefficient than ARMs. Specifically, for a given batch, ARMs will compute loss on each token. MDMs only compute loss on the masked tokens. In a sense, processing all the unmasked tokens is “wasted compute” from a learning perspective.

### **Other minor comments / suggestions**

1. Throughout, when citing MDLM (Sahoo et al. [3]) I would recommend citing the concurrent works of Shi et al. [4] and Ou et al. [5].
2. Additionally, there seems to be a repeated entry in the references for Sahoo et al. [3]
3. (Feel free to ignore this but,) I think clarity of exposition in the intro could be improved by removing the mathematical notation for the joint distribution factorization of MDMs and ARMs, which is described in more detail in Section 3. Additionally, there is no need to include “($R_D^*$)” in line 57; this term is not re-used or explained here.
4. End of line 267 has a typo: “very less decay”

---

[1] Yang, An, et al. "Qwen3 technical report." arXiv preprint arXiv:2505.09388 (2025).

[2] Nie, Shen, et al. "Large language diffusion models." arXiv preprint arXiv:2502.09992 (2025).

[3] Sahoo, Subham, et al. "Simple and effective masked diffusion language models." Advances in Neural Information Processing Systems 37 (2024): 130136-130184.

[4] Shi, Jiaxin, et al. "Simplified and generalized masked diffusion for discrete data." Advances in neural information processing systems 37 (2024): 103131-103167.

[5] Ou, Jingyang, et al. "Your absorbing discrete diffusion secretly models the conditional distributions of clean data." arXiv preprint arXiv:2406.03736 (2024).

---

> ### Author Rebuttal · Authors · 2025-07-31
>
> Thank you for your constructive feedback and for carefully reviewing our work.
>
> Below, we address each of your points in turn:
>
> ---
>
> ### **Q4.1: Further experiments validating the hypothesis on *why* MDMs are more data-efficient.**
>
> To better understand why MDMs outperform ARMs in low-data regimes, we conducted a series of ablations on ARM models to test whether exposing them to more diverse prediction tasks, which is similar to what MDMs inherently do, could close the gap or interpolate between the two methods.
>
> We hypothesized in the paper:
> > *“MDM is exposed to diverse token orderings and prediction tasks, unlike the fixed left-to-right training of AR models. This enables them to extract more information per example.”*
>
> To validate this, we conducted the following experiments:
>
> #### **(a) Random Token Masking in ARM**
>
> | Token Mask Ratio | Val Loss |
> |------------------|----------|
> | 0% (Default)     | **3.71630**  |
> | 12.5%            | 3.72701  |
> | 25%              | 3.75580  |
> | 37.5%            | 3.81153  |
> | 50%              | 3.90185  |
>
> Token masking generally degrades validation loss, likely due to over-regularization.
>
> #### **(b) Attention Dropout in ARM**
>
> | Dropout %        | Val Loss |
> |------------------|----------|
> | 10% (Default)    | 3.71630  |
> | 25%              | **3.71268**  |
> | 50%              | 3.72415  |
> | 75%              | 3.76850  |
>
> Moderate dropout provides marginal gains but degrades performance at higher levels.
>
> #### **(c) Randomizing Token Order (Sigma-GPT [1] / RandAR [2])**
> In order to show the full impact, including the impact on overfitting points, we used a setting of 100 epochs here. Therefore, the original result for AR is different from the previous two experiments (50 epochs).
>
> | Variant              | Val Loss |
> |----------------------|----------|
> | Original | 3.74799  |
> | 25% Random Order     | 3.67741  |
> | 50% Random Order     | **3.66556**  |
>
> Randomized token ordering shows the most promising improvement, supporting our hypothesis that MDMs benefit from diverse prediction tasks. We implemented RandAR (Any-Order AR) on top of our ARM baseline for this experiment.
>
> ---
>
> ### **Q4.2: Motivation of internet-scale data is not aligned with small-scale experiments.**
>
> We agree that current LLMs are trained on trillions of tokens, which is larger than the data regimes we study. However, we believe our results remain meaningful:
> - Our experiments demonstrate that objectives inducing randomized ordering (MDM) are more data-efficient.
> - This insight can motivate future hybrid approaches, where diffusion-like objectives are applied on scarce high-quality data, while AR is applied on abundant lower-quality data.
> - Domains with inherently limited data (e.g., robotics and healthcare) would particularly benefit from MDM-like data efficiency.
>
> Since submission, we have scaled our experiments to **500M unique tokens** using our critical compute scaling law (Appendix C), We train:
>
> - 2.3B MDM trained for 130 epochs (terminated due to compute limits)
> - Best ARM configuration via grid search for the same data budget
>
> We report the results below:
>
> **Validation Losses:**
>
> - **MDM:** 3.08573
> - **ARM:** 3.20136
>
> ---
>
> ### **Q4.3: Authors should discuss Llada [2], which shows more comparable scaling laws using downstream metrics.**
>
> Thank you for pointing this out. We will include a discussion of Llada [2] in the final version. Our scaling law analysis focused on validation loss, which is standard in scaling law studies and provides a stable signal.
>
> To complement this, we conducted additional **downstream evaluations**, which confirm that MDM’s lower validation loss translates to better generalization:
>
> **Table 2: Accuracy (%) on downstream evaluation tasks at different unique token scales.**
>
> | Benchmark       | Random | ARM (100M) | MDM (100M) | ARM (500M) | MDM (500M) |
> |-----------------|--------|------------|------------|-------------|-------------|
> | ARC-Easy        | 25.00  | 35.63      | **37.84**  | 43.79       | **45.95**   |
> | BoolQ           | 50.00  | 46.00      | **49.38**  | 51.87       | **55.26**   |
> | COPA            | 50.00  | 56.33      | **59.00**  | **67.00**   | 64.83       |
> | HellaSwag       | 25.00  | 27.37      | **30.24**  | 32.28       | **35.33**   |
> | PiQA            | 50.00  | **60.94**      | 60.72  | **65.71**   | 65.61       |
> | RACE            | 25.00  | 25.28      | **28.96**  | 28.28       | **31.44**   |
> | WinoGrande XL   | 50.00  | 48.87      | **50.97**  | 50.61       | **51.51**   |
> | SciQ            | 25.00  | 58.05      | **68.67**  | 67.82       | **79.13**   |
> | Lambda          | 00.00  | 10.91      | **15.19**  | 15.07       | **22.30**   |
>
> *Note: Best results shown in bold.*
>
> ---
>
> ### **Q4.4: Disagreement with lines 202–205 claiming MDMs are more compute-inefficient than ARMs.**
>
> We appreciate the clarification. We **agree** that MDM computes loss on only the masked tokens, which provides less direct supervision per update and higher gradient variance than ARM, where every token is supervised.
>
> However, we **disagree** that processing unmasked tokens is “wasted compute”:
> 1. Unmasked tokens contribute to the loss indirectly via gradient propagation.
> 2. Unmasked inputs act as registers, gathering and passing information that improves predictions at masked positions.
>
> Our conclusion that MDMs are more compute-inefficient is empirically grounded: achieving the same validation loss as ARM requires more FLOPs as a result more GPU-hours, as is also shown in prior works [16, 24].  .
> Future work may mitigate this, e.g., Arriola et al. [1] show that tuning the masking schedule can reduce gradient variance and improve compute efficiency. We will add further clarification in the final version of the paper on this.
>
> ---
> ### **Q4.5: Other Minor Comments /Suggestions**
>
> Thank you for these helpful suggestions. We will incorporate the fixes in the final version of the paper
>
> ---
>
> ### **Q4.6: Difference between Table 1a and Table 1b? Why is the R² so poor in Table 1b?**
>
> - **Table 1a** represents scaling law fitting in the **single-epoch setting** (Chinchilla-style compute/data scaling).
> - **Table 1b** represents scaling law fitting in the **multi-epoch, data-constrained setting** (Muennighoff et al. [15]) and estimates the constants \(R_D^*\) and \(R_N^*\).
>
> The initial submission showed lower R² in Table 1b due to the inclusion of:
> 1. Configurations with <20 epochs for MDMs
> 2. ARM models with <45M parameters
>
> These configurations produced high-variance validation loss, making regression noisier.
>
> Since submission, we have refit the scaling laws, excluding these high-variance points, resulting in more stable and interpretable fits:
>
> **Updated Table 1:**
>
> **(a) Initial fit**
>
> | Model | R²     | Loss       |
> |-------|--------|------------|
> | MDM   | 0.9447 | 0.0002     |
> | ARM   | 0.9439 | 7.7532e−05 |
>
> **(b) Second-step fit with extracted scaling parameters**
>
> | Model | R²     | Loss     | R\*_D  | R\*_N   |
> |-------|--------|----------|--------|---------|
> | MDM   | 0.9784 | 0.00079  | 493.89 | 1265.65 |
> | ARM   | 0.7628 | 0.00361  | 31.19  | 55.16   |
>
> ---
>
> ### **Q4.7: In Figure 2, isn’t the NLL relatively poor compared to Table 2 in MDLM [3]?**
>
> The comparison is not directly fair because:
> - MDLM Table 2 is trained on **262B unique tokens**, while our experiments use **100M unique tokens**.
> - If we assume 100 epochs of repetition is the same as unique tokens ≈ 10B effective tokens, this is still **26× smaller** than MDLM.
>
> Further we would like to point, our best ARM validation loss (3.71) in the 100M regime (Figure 2) closely matches 3.72 reported by Muennighoff et al. for the same scale (Figure 3 of their paper).
>
> ---
>
> ### **Q4.8: Authors are missing a scaling factor for the MDM loss.**
>
> Thank you for pointing it out. Yes, this is a typo and will be fixed in the final version.
>
> ---
>
> ### **Q4.9:  There is no additional pre-training loss to enable CFG in Nie et al. [6]?**
>
> Correct. We acknowledge this and will remove the line in the final version. Thanks for pointing it out!
>
> ---
>
> ### **Q4.10: In Figure 2, is the colorful “Chinchilla-optimal” star for 1 epoch of training?**
>
> Yes, the star corresponds to 1 epoch of training under Chinchilla-optimal scaling.
>
> ---
>
> ### **Q4.11: Why are there different model sizes in the legends of Figure 5 for ARM vs. MDM?**
>
> We fixed the training compute to \(1 \times 10^{19}\), \(3 \times 10^{19}\), and \(1 \times 10^{20}\) FLOPs.
> We then used the compute-optimal scaling law to determine model sizes for each family:
> - For \(1 \times 10^{19}\) FLOPs → ARM ≈ 217M, MDM ≈ 117M
>
> Hence, model sizes differ because compute-optimal configurations are different for ARMs vs. MDMs.
>
> ---
>
> ### **Q4.12: What does “showing masking, not attention, is key to their advantage when data is scarce” mean?**
>
> We agree that this sentence was unclear and will remove it in the final version.
> Our updated experiments (see Q4.1) clarify that **randomized token orderings** are the main driver of MDM’s data efficiency.
>
>
>
>
>
> **References:**
>
> [1] Arriola, E., et al. *Improving compute efficiency of diffusion models via masking schedules.*
> [2] Llada, J., et al. *Downstream scaling laws for generative models.*

---

> > ### Comment · Reviewer_5KGy · 2025-08-01
> >
> > Thank you for the very detailed response and clarifications
> >
> > I believe the new experiments on the ablated AR model are interesting and the inclusion of downstream tasks is useful.
> >
> > I have raised my score to "5: accept" accordingly.

---

### Official Review · Reviewer_43Rm · 2025-06-30

**Clarity:** 2
**Significance:** 3
**Originality:** 3
**Rating:** 4
**Confidence:** 2

**Summary:**

This paper empirically demonstrates that masked diffusion models (MDMs) can outperform autoregressive models (ARMs) in data-constrained settings. This finding offers a new perspective because most existing large language models are based on ARMs. Inspired by the observation that ARMs become less effective with more training epochs, the authors address the problem of performance in data-limited scenarios. They compare ARMs and MDMs using the same architecture, with various data and model size. The results show that, given sufficient compute in multi-epoch training, MDMs achieve lower validation loss and better downstream performance compared to ARMs.

**Questions:**

* Related to the first point in the Weaknesses, it would be helpful if they provide empirical results showing that MDMs outperform ARMs in data-constrained settings with other datasets or model architectures.

* Related to the second point in the Weaknesses, it would be valuable to see if MDMs remain effective when the ARM's optimizer settings are adjusted.

* Related to the third point in the Weaknesses, it would be useful to verify whether MDMs demonstrate effective performance on tasks that can be evaluated by human judgment.

**Ethical Concerns:**

["NO or VERY MINOR ethics concerns only"]

**Final Justification:**

Through the authors' response and additional experiments, I believe this paper provides a valuable discussion point on one of the key ongoing topics in the language modeling community: the comparison between autoregressive and diffusion models.

However, since the primary focus of this work is on empirical comparison, I find it difficult to expect fundamental insights into the underlying aspects of the models. Therefore, I believe that my current borderline accept recommendation remains appropriate.

**Limitations:**

Yes.

**Quality:**

2

**Strengths And Weaknesses:**

## [Strengths]

* Data-constrained settings are important not only for domains where language models must be trained with limited data, but also for cases that may arise even in large language models. Therefore, this work addresses a highly significant problem.

* They conduct experiments on hundreds of models and provide an evaluation across various data sizes, model scales, and numbers of training epochs.

## [Weaknesses]
* Although the authors conducted experiments with various model and dataset sizes, there are still some limitations in the experimental setup. The experiments seem to focus on the English C4 dataset and the GPT-style Transformer model. While I appreciate that the paper presents results for a range of model and data sizes, it would provide stronger evidence if similar results could be demonstrated on other datasets and model architectures.

* Since the losses for ARMs and MDMs differ, the optimal optimizer settings (such as learning rate, optimizer choice, etc.) may differ as well. However, it appears that the same settings are used for both. This raises the question of whether the chosen hyperparameters might be particularly well-suited for MDMs. I believe this point warrants further discussion.

* While the experimental results are presented primarily in terms of NLL-based loss values, it is also important to evaluate the effectiveness of the generated text outputs through human judgment or downstream tasks that reflect real-world utility.

---

> ### Author Rebuttal · Authors · 2025-07-31
>
> Thank you for your constructive feedback and for carefully reviewing our work.
>
> Below, we address your points in turn:
>
> ---
>
> ### **Q3.1: Extensions to other datasets and model architectures.**
>
> Thank you for this suggestion. We agree that validating our findings across different datasets and model architectures will strengthen the paper.
> We are **currently running experiments on SlimPajama** (a large, high-quality text dataset) and on **T5 (encoder-decoder architecture)**.   These experiments are ongoing, and we plan to include the results in the discussion phase or in the final version of the paper.
>
> ---
>
> ###  **Q3.2: Varying optimizer settings for ARM. Chose hyperparameters might be particularly well-suited for MDMs**
>
> We followed the exact hyperparameter settings from the NeurIPS data-constrained Scaling Law paper (Muennighoff et al. [15]), which uses hyperparameters commonly used for training autoregressive language models. We believe these hyperparameters may slightly favor ARMs, as these were originally tuned for that family.
>
> Importantly:
> - Our **best ARM validation loss** in the 100M unique token setting is **3.71** (Figure 2 of our paper), which closely matches the **3.72** reported by Muennighoff et al. (Figure 3 of their paper).
> - Our **MDM achieves a validation loss of 3.55**, which is unlikely to be matched by ARM even with hyperparameter tweaks.
>
> That said, we are currently running additional experiments with varied learning rates for ARM to confirm this point and will include results in the discussion/final version.
>
> ---
>
> ###  **Q3.3: More downstream experiments.**
>
> To assess whether MDM’s lower validation loss translates to better generalization, we evaluated the **best-performing MDM and ARM models** on a range of language understanding tasks.  Across all tasks and data scales, MDM consistently outperforms ARM.
>
> **Table 2: Accuracy (%) on downstream evaluation tasks at different unique token scales.**
>
> | Benchmark       | Random | ARM (100M) | MDM (100M) | ARM (500M) | MDM (500M) |
> |-----------------|--------|------------|------------|-------------|-------------|
> | ARC-Easy        | 25.00  | 35.63      | **37.84**  | 43.79       | **45.95**   |
> | BoolQ           | 50.00  | 46.00      | **49.38**  | 51.87       | **55.26**   |
> | COPA            | 50.00  | 56.33      | **59.00**  | **67.00**   | 64.83       |
> | HellaSwag       | 25.00  | 27.37      | **30.24**  | 32.28       | **35.33**   |
> | PiQA            | 50.00  | **60.94**   |  60.72  | **65.71**   | 65.61       |
> | RACE            | 25.00  | 25.28      | **28.96**  | 28.28       | **31.44**   |
> | WinoGrande XL   | 50.00  | 48.87      | **50.97**  | 50.61       | **51.51**   |
> | SciQ            | 25.00  | 58.05      | **68.67**  | 67.82       | **79.13**   |
> | Lambda          | 00.00  | 10.91      | **15.19**  | 15.07       | **22.30**   |
>
> *Note: Best results shown in bold.*
>
> ---
>
> ## **Additional Updates**
> We have also conducted additional experiments since our submission. We summarize these updates below:
>
> ---
>
> ### **i) Why is MDM Better in Data-Constrained Settings?**
>
> To better understand why MDMs outperform ARMs in low-data regimes, we conducted a series of ablations on ARM models to test whether exposing them to more diverse prediction tasks, which is similar to what MDMs inherently do, could close the gap or interpolate between the two methods.
>
> We hypothesized in the paper:
> > *“MDM is exposed to diverse token orderings and prediction tasks, unlike the fixed left-to-right training of AR models. This enables them to extract more information per example.”*
>
> To validate this, we conducted the following experiments:
>
> #### **(a) Random Token Masking in ARM**
>
> | Token Mask Ratio | Val Loss |
> |------------------|----------|
> | 0% (Default)     | **3.71630**  |
> | 12.5%            | 3.72701  |
> | 25%              | 3.75580  |
> | 37.5%            | 3.81153  |
> | 50%              | 3.90185  |
>
> We observe that token masking in AR degrades validation loss, especially at higher ratios, likely due to over-regularization.
>
> #### **(b) Attention Dropout in ARM**
>
> | Dropout %        | Val Loss |
> |------------------|----------|
> | 10% (Default)    | 3.71630  |
> | 25%              | **3.71268**  |
> | 50%              | 3.72415  |
> | 75%              | 3.76850  |
>
> Moderate dropout yields marginal improvements but leads to degradation at higher levels.
>
> #### **(c) Randomizing Token Order (Sigma-GPT [1] / RandAR [2])**
> In order to show the full impact, including the impact on overfitting points, we used a setting of 100 epochs here. Therefore, the original result for AR is different from the previous two experiments (50 epochs).
>
> | Variant              | Val Loss |
> |----------------------|----------|
> | Original  | 3.74799  |
> | 25% Random Order     | 3.67741  |
> | 50% Random Order     | **3.66556**  |
>
> Randomized orderings show the most promising gains, supporting our hypothesis that MDMs benefit from diverse prediction structures. For this experiment, we implemented RandAR (Any-Order AR) on top of our ARM baseline.
>
> ---
>
> ### **ii) Scaling to 500M Unique Tokens Using the Critical Compute Equation**
>
> In Appendix C, we derive an analytical equation that estimates the critical compute at which MDM matches the validation loss of ARM, given a fixed unique data budget. Using this, we scaled the training set to 500M unique tokens and performed a grid search over model sizes and total data based on the predicted compute to identify the configuration predicted by the scaling law to achieve the optimal validation loss as close as possible. Therefore, we trained a 2.3B parameter MDM using the predicted critical compute budget. After training for 130 epochs, we observed that the validation loss of MDM is consistently lower than that of ARM.
>
> For the ARM baseline, we use the same method to find out optimal setting under the same data budget. We report the empirical validation losses of the trained models below:
>
> **Validation Losses:**
>
> - **MDM:** 3.08573
> - **ARM:** 3.20136
>
> ---
>
> ### **iii) Improved Fitting Metrics (in Appendix)**
>
> We refit the scaling laws after removing configurations with a small number of training epochs (<20) for MDMs and excluding very small model sizes (<45M parameters) for ARMs. These configurations often exhibited high variance in validation loss due to insufficient optimization, introducing noise into the regression. Excluding them resulted in significantly more stable and interpretable fits.
>
> **Updated Table 1:**
>
> **(a) Initial fit**
>
> | Model | R²     | Loss       |
> |-------|--------|------------|
> | MDM   | 0.9447 | 0.0002     |
> | ARM   | 0.9439 | 7.7532e−05 |
>
> **(b) Second-step fit with extracted scaling parameters**
>
> | Model | R²     | Loss     | R\*_D  | R\*_N   |
> |-------|--------|----------|--------|---------|
> | MDM   | 0.9784 | 0.00079  | 493.89 | 1265.65 |
> | ARM   | 0.7628 | 0.00361  | 31.19  | 55.16   |
>
> ---
>
> **References:**
>
> [1] Pannatier, Arnaud, Evann Courdier, and François Fleuret. *σ-GPTs: A new approach to autoregressive models.*
>
> [2] Pang, Ziqi, et al. *RandAR: Decoder-only autoregressive visual generation in random orders.*

---

> > ### Comment · Reviewer_43Rm · 2025-08-04
> >
> > Thank you to the authors for their response and additional experiments. I believe these further experiments strengthen the validity of the paper. I will maintain my current recommendation toward acceptance.

---

> ### Author Response · Authors · 2025-08-05
> **Additional Experiments**
>
> Dear Reviewer,
>
> We ran additional experiments as per your suggestions:
>
> ### **1. SlimPajama Dataset (100M Unique Tokens)**
>
> To test the generality of our findings, we ran experiments on **SlimPajama**, which is more compressed and information dense than C4, therefore smaller models get preferred.
> We searched around the optimal ARM and MDM settings based on the C4 dataset, varying both model size and number of training epochs.
>
> #### **ARM Sweep (7 configurations):**
> | Model Size | Epochs | Val Loss |
> |------------|--------|----------|
> | 74M        | 100    | **3.44890** *(Best ARM)*
> | 74M        | 200    | 3.45922
> | 140M       | 40     | 3.46200
> | 140M       | 50     | 3.45851
> | 140M       | 100    | 3.52915
> | 217M       | 40     | 3.46917
> | 217M       | 50     | 3.50414
>
> #### **MDM Sweep (Only 2 configurations):**
> | Model Size | Epochs | Val Loss |
> |------------|--------|----------|
> | 425M       | 500    | 3.43250
> | 425M       | 400    | **3.41454** *(Best overall)*
>
> Even though we tried 7 configurations for ARM and only 2 for MDM, we find MDM **outperforms the best ARM by a clear margin**. This reinforces our core finding that MDMs do better in data-constrained settings.
> We expect even stronger MDM performance if a full sweep is conducted on SlimPajama.
>
> ---
>
> ### **2. Optimizer Sensitivity for ARM**
>
> We explored several learning rate and optimizer settings for ARM to check if MDM’s advantage could be attributed to suboptimal AR hyperparameters. Below are the results:
>
> | Config Name | Learning Rate | Min LR | Beta2 | Val Loss |
> |-------------|---------------|--------|-------|----------|
> | Default     | 2e-4          | 2e-5   | 0.95  | **3.72791**
> | Setting 1        | 1e-4          | 1e-5   | 0.95  | 3.80986
> | Setting 2        | 2e-4          | 2e-5   | 0.99  | 3.75094
> | Setting 3        | 3e-4          | 3e-5   | 0.95  | 3.74984
>
> Across all configurations, we find that our **default setup continues to yield the best validation loss**. We believe this is because the optimizer settings we adopted - based on [1] - were already well-tuned for training ARMs under data-constrained settings.
>
> [1] *Muennighoff et al., "Scaling Data-Constrained Language Models", NeurIPS 2023.*

---

> > ### Comment · Reviewer_43Rm · 2025-08-05
> >
> > I appreciate the additional experiments provided in response to my suggestions. They enhance the strength of the paper, and I would be glad to see them included in the final version.

---

### Official Review · Reviewer_u7Lw · 2025-07-01

**Clarity:** 3
**Significance:** 2
**Originality:** 2
**Rating:** 4
**Confidence:** 4

**Summary:**

This paper studies an interesting and timely research question---the scaling law of masked diffusion language models, particularly under data-constrained settings, or equivalently, the setting where multiple iterations over a fixed-size training set are performed. Notably, several core observations have been made, including, for example, the change in validation loss with respect to the number of tokens seen and model size. Experiments also reveal that AR models are more susceptible to overfitting under repeated data points. Overall, evidence show that masked diffusion models are more data-efficient over AR models under certain conditions.

**Questions:**

Q1. The formulations in line 27 and line 32 for AR and MDMs are not rigorously correct. While I can get the point that the authors want to highlight the difference between AR and MDMs in terms of ordering, I would not recommend loosely writing the factorizations in this way. As already said in Weakness, the AR factorization in line 27 is not exactly true. the MDM factorization in line 32 is not rigor as well, and distinctions should be made between MDMs and any-order ARs [1] to avoid confusions.

Q2. It has been revealed that MDMs require larger number of epochs over limited training data to achieve the peak performance. Should other techniques that can possibly lead to different training dynamics (like higher learning rates) help mitigate the issue?

Q3. The conclusion drawn from this paper indicates that MDMs could outperform ARs when training set is limited. However, this is in contrary to the large-scale pretraining paradigm adopted for modern LLMs. Does this research convey a pessimistic signal that MDMs are not suitable / may not be able to outperform / replace ARs as a new generation of large language models?

Q4. There is an important factor which I believe requires further elaboration. While the paper has clearly distinguished number of unique tokens and number of total tokens used during training, for MDMs there is another critical concept of "effective tokens", i.e., the tokens that are used to perform gradient descent (those that are masked in each interation) that also leads to the difference of training dynamics between MDMs and ARs. In particular, in each iteration ARs can perform gradient update using all tokens while MDMs can only use effectively half of them, assuming a uniform mask rate between 0 and 1. This concept has been mentioned in previous works such as [2] but is never discussed and analyzed in this paper.

[1] Shih et al. Training and Inference on Any-Order Autoregressive Models the Right Way. NeurIPS 2022.

[2] Arriola et al. Block diffusion: Interpolating between autoregressive and diffusion language models. ICLR 2025.

**Ethical Concerns:**

["NO or VERY MINOR ethics concerns only"]

**Final Justification:**

With the rebuttal, the paper is in much better shape with sufficient results to support the claims, as an empirical research. The authors are recommended to incorporate the promised changes to the manuscript.

**Limitations:**

yes

**Quality:**

2

**Strengths And Weaknesses:**

## Strengths

1. The paper is well-motivated and the study of the scaling law of masked diffusion model is a timely research for the community.

2. The aspects studied in this paper are comprehensive, including many interesting and detailed investigations.

## Weaknesses

1. Regarding technical novelty, the methodology basically follows previous investigations of the scaling law for AR language models, yielding limited technical contribution.

2. As an empirical study, it would be better, if more profound and deeper understanding about not only "what" the observations are, but also "why" such things happen, can be discussed and analyzed. However, these insights are absent in the current content.

3. Several statements made in the paper lack rigor. For instance, line 27 the factorization did not specify the meaning of $x_i$. And if they refers to tokens, AR model actually factorizes $p(x)=p(x_0)p(x_1|x_0)p(x_2|x_1,x_0)$, which is different from the formulation presented in this paper. See Questions for more details.

---

> ### Author Rebuttal · Authors · 2025-07-31
>
> Thank you for your constructive feedback and thoughtful suggestions.
>
> Below, we address each of your points in turn:
>
> ---
>
> ### **Q2.1: Limited technical novelty.**
>
> We agree that our work does not introduce a new algorithm. However, we believe our empirical findings are novel and important for the generative modeling community. Our work is the first to systematically characterize the **data vs. compute trade-offs** between MDMs and ARMs across a wide range of model and data scales, and to establish a **closed-form scaling law** for the critical compute threshold at which MDMs outperform ARMs.
>
> These insights, while empirical, can provide a foundation for future algorithms that can better interpolate between the compute-efficiency of ARMs and the data-efficiency of MDMs.
>
> ---
>
> ### **Q2.2: More analysis on *why* diffusion is better in data-constrained settings.**
>
> To better understand why MDMs outperform ARMs in low-data regimes, we conducted a series of ablations on ARM models to test whether exposing them to more diverse prediction tasks, which is similar to what MDMs inherently do, could close the gap or interpolate between the two methods.
>
> We hypothesized in the paper:
> > *“MDM is exposed to diverse token orderings and prediction tasks, unlike the fixed left-to-right training of AR models. This enables them to extract more information per example.”*
>
> To validate this, we conducted the following experiments:
>
> #### **(a) Random Token Masking in ARM**
>
> | Token Mask Ratio | Val Loss |
> |------------------|----------|
> | 0% (Default)     | **3.71630**  |
> | 12.5%            | 3.72701  |
> | 25%              | 3.75580  |
> | 37.5%            | 3.81153  |
> | 50%              | 3.90185  |
>
> We observe that token masking in AR degrades validation loss, likely due to over-regularization.
>
> #### **(b) Attention Dropout in ARM**
>
> | Dropout %        | Val Loss |
> |------------------|----------|
> | 10% (Default)    | 3.71630  |
> | 25%              | **3.71268**  |
> | 50%              | 3.72415  |
> | 75%              | 3.76850  |
>
> Moderate dropout yields minor improvements, but quickly becomes harmful at higher levels.
>
> #### **(c) Randomizing Token Order (Sigma-GPT [1] / RandAR [2])**
>
> | Variant              | Val Loss |
> |----------------------|----------|
> | Original | 3.74799  |
> | 25% Random Order     | 3.67741  |
> | 50% Random Order     | **3.66556**  |
>
> Randomized token orderings show the most promising improvements, supporting our hypothesis that MDM’s exposure to diverse prediction structures enables greater information extraction per example. For this experiment, we implemented RandAR (Any-Order AR) on top of our ARM baseline.
>
> ---
>
> ### **Q2.3: Factorizations for AR and MDM in Lines 27 and 32 lack rigor.**
>
> Thank you for pointing out. We agree that the factorizations for both AR and MDM were oversimplified and may be misleading, and will revise the text to refer readers directly to the method section, where the model factorizations are explained rigorously and unambiguously.
>
> In particular:
> - For AR models, we will clarify that the joint distribution is factorized as:
> $$ p(x) = \prod_{t=0}^{T-1} p(x_t \mid x_{<t}) $$
>  , rather than the simplified and potentially misleading Markovian example currently shown.
>
> - For MDM, we acknowledge that writing the joint distribution in a factorized form is not theoretically grounded. Unlike AR or any-order AR models, MDMs do not explicitly define a joint factorization. To avoid confusion, we will remove the equation in Line 32 and use the following sections to explain in detail.
>
> ---
>
> ### **Q2.4: Can higher learning rates mitigate MDM’s compute inefficiency?**
>
> We do not expect this to be the case. Multiple prior works have noted that **MDMs require more compute** than ARMs to reach similar validation losses (e.g., [1, 16, 24]). We believe these inefficiencies arise not from optimization difficulties, but from fundamental differences in supervision density and variance in gradient (see Q2.6).
>
> That said, we are currently running additional experiments with higher learning rates to test whether optimization plays any non-negligible role. We will include updated results in the discussion/final version once available.
>
> ---
>
> ### **Q2.5: Does this research convey a pessimistic signal that MDMs are not suitable as the next generation of large language models?**
>
> We do not believe so. As Ilya Sutskever aptly said, *"Compute is growing at a rapid pace, but data is not—we have just one internet."* Our findings suggest that MDMs are significantly more data-efficient than ARMs, which may make them better suited for future regimes where high-quality unique data is scarce.
>
> We envision a number of exciting research directions that build on our work:
>
> - **Interpolating between MDM and ARM objectives** to achieve better trade-offs between compute and data efficiency.
> - **Modality- or quality-adaptive objective**, where MDM is used on scarce, high-quality data and ARM is used on abundant, low-quality data.
> - Applications in domains such as **robotics and healthcare**, where one doesn't have billions/trillions of tokens to begin with and data-efficient generative models are critical.
>
> Rather than pessimistic, we view our work as an invitation to rethink model design in the post-internet era.
>
> ---
>
> ### **Q2.6: Further elaboration on effective tokens seen by MDM.**
>
> Thank you for highlighting this. We agree that in MDM training, loss is computed on only a subset of tokens (in expectation, 50%), compared to ARM where every token is supervised. We believe this could contribute to the compute inefficiency of MDMs and have added the following discussion in our paper:
>
> > **Why are autoregressive (AR) models more compute-efficient than diffusion models?**
> > We hypothesize the following as contributing factors:
> > (i) *Stronger supervision per update:* In AR training, every token in a sequence is a supervised target, and the causal dependency structure enables dense, low-variance gradient updates.
> > (ii) *Sparser supervision in MDM:* In contrast, MDM loss is computed on a subset of masked tokens, and while gradients propagate through the entire sequence, each update carries less direct learning signal.
> > Arriola et al. [1] demonstrate that tuning the masking schedule can help reduce gradient variance and improve compute efficiency in MDMs.
>
> We thank the reviewer again for this useful clarification.
>
> ---
> ## **Additional Updates**
>
> We have also conducted additional experiments since our submission. We summarize these updates below:
>
> ---
>
> ### **i) Scaling to 500M Unique Tokens Using the Critical Compute Equation**
>
> In Appendix C, we derive an analytical equation that estimates the critical compute at which MDM matches the validation loss of ARM, given a fixed unique data budget. Using this, we scaled the training set to 500M unique tokens and performed a grid search over model sizes and total data based on the predicted compute to identify the configuration predicted by the scaling law to achieve the optimal validation loss as close as possible. Therefore, we trained a 2.3B parameter MDM using the predicted critical compute budget. After training for 130 epochs, we observed that the validation loss of MDM is consistently lower than that of ARM.
>
> For the ARM baseline, we use the same method to find out optimal setting under the same data budget. We report the empirical validation losses of the trained models below:
>
> **Validation Losses:**
>
> - **MDM:** 3.08573
> - **ARM:** 3.20136
>
> ---
>
> ### **ii) More Downstream Results**
>
> We further evaluated the best-performing MDM and ARM models on a range of language understanding tasks to assess whether the gains in validation loss translate to better generalization. Across a diverse set of tasks and data scales, MDMs consistently outperform ARMs.
>
> **Table 2: Accuracy (%) on downstream evaluation tasks at different unique token scales.**
>
> | Benchmark       | Random | ARM (100M) | MDM (100M) | ARM (500M) | MDM (500M) |
> |-----------------|--------|------------|------------|-------------|-------------|
> | ARC-Easy        | 25.00  | 35.63      | **37.84**  | 43.79       | **45.95**   |
> | BoolQ           | 50.00  | 46.00      | **49.38**  | 51.87       | **55.26**   |
> | COPA            | 50.00  | 56.33      | **59.00**  | **67.00**   | 64.83       |
> | HellaSwag       | 25.00  | 27.37      | **30.24**  | 32.28       | **35.33**   |
> | PiQA            | 50.00  | **60.94**      |  60.72  | **65.71**   | 65.61       |
> | RACE            | 25.00  | 25.28      | **28.96**  | 28.28       | **31.44**   |
> | WinoGrande XL   | 50.00  | 48.87      | **50.97**  | 50.61       | **51.51**   |
> | SciQ            | 25.00  | 58.05      | **68.67**  | 67.82       | **79.13**   |
> | Lambda          | 00.00  | 10.91      | **15.19**  | 15.07       | **22.30**   |
>
> *Note: Best results shown in bold.*
>
> ---
>
> ### **iii) Improved Fitting Metrics (in Appendix)**
>
> We refit the scaling laws after removing configurations with a small number of training epochs (<20) for MDMs and excluding very small model sizes (<45M parameters) for ARMs. These configurations often exhibited high variance in validation loss due to insufficient optimization, introducing noise into the regression. Excluding them resulted in significantly more stable and interpretable fits.
>
> **Updated Table 1:**
>
> **(a) Initial fit**
>
> | Model | R²     | Loss       |
> |-------|--------|------------|
> | MDM   | 0.9447 | 0.0002     |
> | ARM   | 0.9439 | 7.7532e−05 |
>
> **(b) Second-step fit with extracted scaling parameters**
>
> | Model | R²     | Loss     | R\*_D  | R\*_N   |
> |-------|--------|----------|--------|---------|
> | MDM   | 0.9784 | 0.00079  | 493.89 | 1265.65 |
> | ARM   | 0.7628 | 0.00361  | 31.19  | 55.16   |
>
>
> **References:**
>
> [1] Pannatier, Arnaud, Evann Courdier, and François Fleuret. *σ-GPTs: A new approach to autoregressive models.*
>
> [2] Pang, Ziqi, et al. *RandAR: Decoder-only autoregressive visual generation in random orders.*

---

> > ### Author Response · Authors · 2025-08-05
> >
> > Dear Reviewer,
> >
> > We ran additional experiments as per Reviewer 43Rm29’s suggestions, which we would like to share:
> >
> > ### **1. SlimPajama Dataset (100M Unique Tokens)**
> >
> > To test the generality of our findings, we ran experiments on **SlimPajama**, which is more compressed and information dense than C4, therefore smaller models get preferred.
> > We searched around the optimal ARM and MDM settings based on the C4 dataset, varying both model size and number of training epochs.
> >
> > #### **ARM Sweep (7 configurations):**
> > | Model Size | Epochs | Val Loss |
> > |------------|--------|----------|
> > | 74M        | 100    | **3.44890** *(Best ARM)*
> > | 74M        | 200    | 3.45922
> > | 140M       | 40     | 3.46200
> > | 140M       | 50     | 3.45851
> > | 140M       | 100    | 3.52915
> > | 217M       | 40     | 3.46917
> > | 217M       | 50     | 3.50414
> >
> > #### **MDM Sweep (Only 2 configurations):**
> > | Model Size | Epochs | Val Loss |
> > |------------|--------|----------|
> > | 425M       | 500    | 3.43250
> > | 425M       | 400    | **3.41454** *(Best overall)*
> >
> > Even though we tried 7 configurations for ARM and only 2 for MDM, we find MDM **outperforms the best ARM by a clear margin**. This reinforces our core finding that MDMs do better in data-constrained settings.
> > We expect even stronger MDM performance if a full sweep is conducted on SlimPajama.
> >
> > ---
> >
> > ### **2. Optimizer Sensitivity for ARM**
> >
> > We explored several learning rate and optimizer settings for ARM to check if MDM’s advantage could be attributed to suboptimal AR hyperparameters. Below are the results:
> >
> > | Config Name | Learning Rate | Min LR | Beta2 | Val Loss |
> > |-------------|---------------|--------|-------|----------|
> > | Default     | 2e-4          | 2e-5   | 0.95  | **3.72791**
> > | Setting 1        | 1e-4          | 1e-5   | 0.95  | 3.80986
> > | Setting 2        | 2e-4          | 2e-5   | 0.99  | 3.75094
> > | Setting 3        | 3e-4          | 3e-5   | 0.95  | 3.74984
> >
> > Across all configurations, we find that our **default setup continues to yield the best validation loss**. We believe this is because the optimizer settings we adopted - based on [1] - were already well-tuned for training ARMs under data-constrained settings.
> >
> > [1] *Muennighoff et al., "Scaling Data-Constrained Language Models", NeurIPS 2023.*

---

> > > ### Comment · Reviewer_u7Lw · 2025-08-05
> > >
> > > Thank you for the detailed response and additional experiments. I will raise the score to 4. I strongly encourage the authors to incorporate the key points in the rebuttal to the manuscript.

---

### Official Review · Reviewer_WHBh · 2025-07-02

**Clarity:** 3
**Significance:** 3
**Originality:** 4
**Rating:** 5
**Confidence:** 4

**Summary:**

This paper identifies a novel setting in which discrete diffusion models are strictly preferable to autoregressive models: specifically, diffusion models are more desirable than AR models in data-constrained regimes.

The authors conducted a detailed scaling law analyses demonstrating that MDMs leverage repeated data more effectively, achieving superior performance across varying model sizes, data volumes, and computational budgets.

**Questions:**

None

**Ethical Concerns:**

["NO or VERY MINOR ethics concerns only"]

**Final Justification:**

I maintain my score.

**Limitations:**

Yes

**Quality:**

3

**Strengths And Weaknesses:**

**Strengths:**

The authors identify a key strength of masked diffusion models, opening up new avenues where these models are strictly preferable.

**Weaknesses:**

The primary takeaway of the paper is that MDMs offer benefits only when the number of unique tokens is 100M or fewer, which feels somewhat underwhelming.


**Minor:**

> “Previous studies comparing diffusion models and autoregressive (AR) models have predominantly focused on the single-epoch regime [16, 21, 24].”

Reference [21] actually trains AR and MDM in a multi-epoch setting on LM1B and OpenWebText, so it should be dropped from this sentence.


**Final Remarks:**

This paper is clearly above the acceptance threshold and should be accepted.

---

> ### Author Rebuttal · Authors · 2025-07-31
>
> Thank you for your constructive feedback and for spending the time to read our paper in detail.
>
> Below we address your concerns:
>
> ---
>
> ### **Q1.1: The primary takeaway of the paper is that MDMs offer benefits only when the number of unique tokens is 100M or fewer, which feels somewhat underwhelming.**
>
> We respectfully disagree with this characterization. The primary contribution of our work is to demonstrate that diffusion models consistently outperform autoregressive models in data-constrained regimes. To test this, we conducted experiments across a range of unique token budgets—25M, 50M, and 100M—in the original submission.
>
> Since submission, we have **extended our experiments to a 500M unique token setting**, where we again observe that MDM outperforms ARM when both are trained under the same data budget, as shown in the updated results below.
>
> While we agree that it would be valuable to validate these trends at even larger scales (e.g., 10B unique tokens), doing so would require training for trillions of tokens to maintain the epoch count (e.g., 500 epochs × 10B = 5T tokens), which is far beyond our current compute budget. That said, we believe the consistency of our results across multiple unique token scales, including up to 500M, could validate our claims.
>
> ---
>
> ### **Q1.2: Reference [21] actually trains AR and MDM in a multi-epoch setting.**
>
> Thank you for pointing this out. We apologize for the oversight and will correct this in the final version.
>
> ---
>
> ## **Additional Updates**
> We have also conducted additional experiments since our submission. We summarize these updates below:
>
> ---
>
> ### **i) Why is MDM Better in Data-Constrained Settings?**
>
> To better understand *why* MDMs outperform ARMs in low-data regimes, we conducted a series of ablations on ARM models to test whether exposing them to more diverse prediction tasks, which is similar to what MDMs inherently do, could close the gap or interpolate between the two methods.
>
> We hypothesized in the paper:
> > *“MDM is exposed to diverse token orderings and prediction tasks, unlike the fixed left-to-right training of AR models. This enables them to extract more information per example.”*
>
> To validate this, we conducted the following experiments:
>
> #### **(a) Random Token Masking in ARM**
>
> | Token Mask Ratio | Val Loss |
> |------------------|----------|
> | 0% (Default)     | **3.71630**  |
> | 12.5%            | 3.72701  |
> | 25%              | 3.75580  |
> | 37.5%            | 3.81153  |
> | 50%              | 3.90185  |
>
> We observe that token masking in AR degrades validation loss, especially at higher ratios, likely due to over-regularization.
>
> #### **(b) Attention Dropout in ARM**
>
> | Dropout %        | Val Loss |
> |------------------|----------|
> | 10% (Default)    | 3.71630  |
> | 25%              | **3.71268**  |
> | 50%              | 3.72415  |
> | 75%              | 3.76850  |
>
> Moderate dropout yields marginal improvements but leads to degradation at higher levels.
>
> #### **(c) Randomizing Token Order (Sigma-GPT [1] / RandAR [2])**
> In order to show the full impact, including the impact on overfitting points, we used a setting of 100 epochs here. Therefore, the original result for AR is different from the previous two experiments (50 epochs).
>
> | Variant              | Val Loss |
> |----------------------|----------|
> | Original | 3.74799  |
> | 25% Random Order     | 3.67741  |
> | 50% Random Order     | **3.66556**  |
>
> Randomized orderings show the most promising gains, supporting our hypothesis that MDMs benefit from diverse prediction structures. For this experiment, we implemented RandAR (Any-Order AR) on top of our ARM baseline.
>
> ---
>
> ### **ii) Scaling to 500M Unique Tokens Using the Critical Compute Equation**
>
> In Appendix C, we derive an analytical equation that estimates the critical compute at which MDM matches the validation loss of ARM, given a fixed unique data budget. Using this, we scaled the training set to 500M unique tokens and performed a grid search over model sizes and total data based on the predicted compute to identify the configuration predicted by the scaling law to achieve the optimal validation loss as close as possible. Therefore, we trained a 2.3B parameter MDM using the predicted critical compute budget. After training for 130 epochs, we observed that the validation loss of MDM is consistently lower than that of ARM.
>
> For the ARM baseline, we use the same method to find out optimal setting under the same data budget. We report the empirical validation losses of the trained models below:
>
> **Validation Losses:**
>
> - **MDM:** 3.08573
> - **ARM:** 3.20136
>
> ---
>
> ### **iii) More Downstream Results**
>
> We further evaluated the best-performing MDM and ARM models on a range of language understanding tasks to assess whether the gains in validation loss translate to better generalization. Across a diverse set of tasks and data scales, MDMs consistently outperform ARMs.
>
> **Table 2: Accuracy (%) on downstream evaluation tasks at different unique token scales.**
>
> | Benchmark       | Random | ARM (100M) | MDM (100M) | ARM (500M) | MDM (500M) |
> |-----------------|--------|------------|------------|-------------|-------------|
> | ARC-Easy        | 25.00  | 35.63      | **37.84**  | 43.79       | **45.95**   |
> | BoolQ           | 50.00  | 46.00      | **49.38**  | 51.87       | **55.26**   |
> | COPA            | 50.00  | 56.33      | **59.00**  | **67.00**   | 64.83       |
> | HellaSwag       | 25.00  | 27.37      | **30.24**  | 32.28       | **35.33**   |
> | PiQA            | 50.00  | **60.94**      | 60.72  | **65.71**   | 65.61       |
> | RACE            | 25.00  | 25.28      | **28.96**  | 28.28       | **31.44**   |
> | WinoGrande XL   | 50.00  | 48.87      | **50.97**  | 50.61       | **51.51**   |
> | SciQ            | 25.00  | 58.05      | **68.67**  | 67.82       | **79.13**   |
> | Lambda          | 00.00  | 10.91      | **15.19**  | 15.07       | **22.30**   |
>
> *Note: Best results shown in bold.*
>
> ---
>
> ### **iv) Improved Fitting Metrics (in Appendix)**
>
> We refit the scaling laws after removing configurations with a small number of training epochs (<20) for MDMs and excluding very small model sizes (<45M parameters) for ARMs. These configurations often exhibited high variance in validation loss due to insufficient optimization, introducing noise into the regression. Excluding them resulted in significantly more stable and interpretable fits.
>
> **Updated Table 1:**
>
> **(a) Initial fit**
>
> | Model | R²     | Loss       |
> |-------|--------|------------|
> | MDM   | 0.9447 | 0.0002     |
> | ARM   | 0.9439 | 7.7532e−05 |
>
> **(b) Second-step fit with extracted scaling parameters**
>
> | Model | R²     | Loss     | R\*_D  | R\*_N   |
> |-------|--------|----------|--------|---------|
> | MDM   | 0.9784 | 0.00079  | 493.89 | 1265.65 |
> | ARM   | 0.7628 | 0.00361  | 31.19  | 55.16   |
>
> ---
>
> **References:**
>
> [1] Pannatier, Arnaud, Evann Courdier, and François Fleuret. *σ-GPTs: A new approach to autoregressive models.*
>
> [2] Pang, Ziqi, et al. *RandAR: Decoder-only autoregressive visual generation in random orders.*

---

### Author Response · Authors · 2025-08-09
**More Experiments to Better Understand "Why" Diffusion Models Outperform Autoregressive Models in Data-Constrained Settings**

Dear Reviewers,

In our prior experiments to understand "why" diffusion is data-efficient, we used RAR baseline, which introduced target positional embeddings, adding extra parameters and potentially conflating the source of improvement. In this new study, we simplified the setup to isolate the effect of *token ordering* alone.

We trained our AR baseline with varying numbers of token orderings:
- **N = 1** → Standard left-to-right ordering
- **N = k** → Left-to-right + *(k − 1)* additional random permutations

We find, as N increases (adding more random permutations), AR models become more data-efficient, showing both improved validation loss and reduced overfitting. This supports our hypothesis that **exposure to diverse token orderings**—inherent to MDM training—is a key factor in their advantage.

| Model / N      | Final Val Loss |
|----------------|----------------|
| Standard AR (N=1) | 4.12538        |
| AR (N=2)      | 3.80137        |
| AR (N=4)      | 3.62011        |
| AR (N=16)     | 3.61096        |
| Best MDM   | 3.55467    |

Experimental details:
- All AR variants trained for 100 epochs on 278m model with 100m unique data
- Evaluation used the standard left-to-right factorization for AR variants.
- The MDM result corresponds to the best loss reported in Figure 2’s contour plot with 100m unique data

These findings provide cleaner evidence that the diverse token orderings in MDMs is central to their improved data efficiency.
Further, our experiments suggest that it may be possible to **interpolate between the compute efficiency of AR and the data efficiency of MDM**. We will include these results in the final version of our paper.

---

### Decision · Program_Chairs · 2025-09-17

**Decision:**

Accept (poster)

**Comment:**

This paper investigates scaling laws for masked diffusion models (MDMs) versus autoregressive models (ARMs) in data-constrained settings, demonstrating that MDMs consistently outperform ARMs when training on limited unique data across multiple epochs. The key scientific finding is that MDMs achieve superior validation loss and downstream task performance when unique token budgets are constrained (25M-500M tokens), with the authors attributing this advantage to MDMs' exposure to diverse token orderings during training. The paper's strengths include comprehensive empirical evaluation across multiple model sizes and data scales, novel identification of a regime where diffusion models are preferable to autoregressive models, and derivation of analytical scaling laws with critical compute thresholds. However, the work has several concerning weaknesses: limited technical novelty, questions about the practical relevance of small-scale "data-constrained" regimes, and more critically, reviewers raised concerns about mathematical rigor in the loss formulations and whether the comparison methodology between AR and diffusion models is fundamentally sound. The validity of using validation loss as the primary comparison metric when the two model types may be measuring fundamentally different quantities (exact likelihood vs. upper bounds) remains questionable and could undermine the paper's central claims.

The rebuttal process addressed many reviewer concerns through extensive additional experiments, though some fundamental methodological questions persist. Reviewer u7Lw raised issues about mathematical formulation rigor, which authors acknowledged by promising to clarify the factorizations, leading to a score maintenance at 4. Reviewer 43Rm's requests for dataset generalization and optimizer sensitivity were addressed with SlimPajama experiments and hyperparameter sweeps. Reviewer 5KGy's concerns about loss formulation details (missing scaling factors, proper lower bounds) were acknowledged as errors to be fixed, and their score increased to 5 after seeing downstream task results. Reviewer WHBh maintained strong support (rating 5) throughout. The paper makes an interesting empirical contribution, but the fundamental validity of the comparison methodology remains a concern and need to be addressed in the future version.